



# A continual learning-based multilayer perceptron for improved reconstruction of three-dimensional nitrate concentration

Xiang Yu[a,b], Huadong Guo[b], Jiahua Zhang[b], Yi Ma[c], Xiaopeng Wang[a], Guangsheng Liu[a], Mingming Xing[b], Nuo Xu[d], and Ayalkibet M. Seka[a,b,e]

[a]Remote Sensing Information and Digital Earth Center, College of Computer Science and Technology, Qingdao University, Qingdao, 266071, China
[b]Aerospace Information Research Institute, Chinese Academy of Sciences, Beijing, 100094, China
[c]First Institute of Oceanography Ministry of National Resource, Qingdao, 266061, China
[d]Department of Biological and Agricultural Engineering, University of California, Davis, 95616, USA
[e]Arba Minch Water Technology Institute, Water Resources Research Center, Arba Minch University, Ethiopia

**Correspondence:** Jiahua Zhang (zhangjh@radi.ac.cn)

**Abstract.** Nitrate plays a crucial role in marine ecosystems, as it influences primary productivity. Despite its ecological significance, accurately mapping its three-dimensional (3D) concentration on a large scale remains a considerable challenge due to the inherent limitations of existing methodologies. To address this issue, this study proposes a continual learning-based multilayer perceptron (MLP) model to reconstruct the 3D ocean nitrate concentrations above 2000 m depth over the pan-European coast.

The continual learning strategy enhances the model generalization by integrating knowledge from CMEMS nitrate data, effectively overcoming the spatial limitations of BGC-Argo observations in comprehensive nitrate characterization. The proposed approach integrates the advantages of extensive spatial remote sensing observations, the precision of Biogeochemical Argo (BGC-Argo) measurements, and the broad knowledge from simulated nitrate datasets, exploiting the capacity of neural networks to model their nonlinear relationships between multi-source sea surface environmental variables and subsurface nitrates.

The model achieves excellent performance in profile cross-validation ($R^2 = 0.98$, RMSE=0.592 $\mu mol \cdot kg^{-1}$), and maintains robustness across diverse 3D validation scenarios, suggesting its effectiveness in filling observational gaps and reconstructing the 3D nitrate field. Then, the spatiotemporal distribution of the reconstructed 3D nitrate field from 2010 to 2023 reveals a spatial distribution pattern, an interannual upward trend, and the degree of consistency in vertical variation. The contributions of all 22 input features to the model's estimation were respectively quantified by using Shapley additive explanations values. This

study reveals the potential of the proposed approach to overcoming observational limitations and enrich further insights into the 3D ocean condition. The reconstructed 3D nitrate dataset is freely available at https://doi.org/10.5281/zenodo.14010813 (Yu et al., 2024).

## 1  Introduction

In the last decade, the global oceans have absorbed approximately 25% of anthropogenic carbon dioxide ($CO_2$) from the at-

mosphere, playing a crucial role in mitigating climate change impacts (Friedlingstein et al., 2020). However, oceanic changes, such as warming and eutrophication may alter this role, leading to complex effects on marine ecosystems and climate. As the



primary limiting nutrient in the upper ocean, nitrate is pivotal in regulating primary productivity, especially new productivity (Bristow et al., 2017; Chen et al., 2023). This could constitute a long-term absorption of CO2 from the surface to the ocean interior (Eppley and Peterson, 1979; Gregg et al., 2003; Joo et al., 2016; Rafter et al., 2017). Thus, a comprehensive comprehension of the temporal and spatial distribution of ocean nitrate is indispensable for conducting research on marine ecology and environment.

Most biogeochemical data are collected in situ via coastal surveillance, oceanographic cruises, offshore platforms or autonomous instruments, such as the Global Ocean Data Analysis Project version 2 database (GLODAPv2) and Biogeochemical Argo (BGC-Argo) (Claustre et al., 2020; Lavigne et al., 2015; Nittis et al., 2007). However, traditional in situ measurements alone cannot provide large-scale and continuous nitrate data. In contrast, remote sensing offers a promising alternative for estimating nitrate due to its broad spatial coverage, temporal consistency, and cost-effectiveness (Chang et al., 2013; Pan et al., 2018). Previous research has successfully utilized it to retrieve water nutrients (Ansper and Alikas, 2019; Du et al., 2020; Mortula et al., 2020; Yu et al., 2016). Machine learning (ML) technologies have also been employed for nutrient concentration retrieval (Huang et al., 2021; Lv et al., 2020; Qun'ou et al., 2021).

Optical satellites face challenges in nitrate retrieval due to the lack of distinctive nitrate signals(Chen et al., 2023; Sathyendranath et al., 1991). Previous studies have demonstrated a strong empirical correlation between SSN and certain measurable seawater parameters (Goes et al., 2000; Joo et al., 2018; Kamykowski et al., 2002; Silió-Calzada et al., 2008; Switzer et al., 2003). Physical processes, biological activity, and chemical reactions like nitrification are commonly recognized as the three principal processes in regulating ocean nitrate (Goes et al., 2000, 1999; Kudela and Dugdale, 2000; Pan et al., 2018). Cold and nitrate-rich water is transported to the euphotic layer through physical processes, including upwelling and convective mixing in winter, enriching SSN while decreasing sea surface temperature (SST) (Kudela and Dugdale, 2000; Pan et al., 2018). Phytoplankton growth consumes nitrate and converts it into organic matter, reducing SSN while increasing Chlorophyll concentration (Chl) (Goes et al., 2000, 1999). Therefore, various physical and biogeochemical characteristics were frequently utilized as features to establish empirical connections with SSN. The conventional method for nitrate retrieval typically relies solely on SST for linear regression, given its negative relationship with SSN (Sarangi and Devi, 2017; Switzer et al., 2003). Nevertheless, the correlation between SST and SSN is subject to significant geographical and temporal variation, influenced by differing environmental conditions across regions (Goes et al., 1999; Silió-Calzada et al., 2008). Goes et al. (1999) found that incorporating Chl-a alongside SST notably improves the accuracy of SSN retrieval compared to using SST in isolation. Additionally, Colored dissolved organic matter (CDOM) is also a feasible candidate for oceans with considerable river inflow (Pan et al., 2018).

One primary limitation of remote sensing retrieval is the challenge of accurately monitoring subsurface environmental parameters (Akbari et al., 2017; Ali et al., 2004). While in situ data provide precise measurements of local vertical conditions, they are inadequate in characterizing ecosystem processes occurring at the extensive temporal and spatial scales involved (Von Schuckmann et al., 2019). Accurate 3D data acquisition for key variables over extensive scales is necessary for a deeper understanding of marine ecosystems (Rossi et al., 2021). To address this issue, various methods including modeling ecosystems and ocean dynamics have been explored to estimate biogeochemical variables, with some being widely applied (Baretta





et al., 1995; Bruggeman and Bolding, 2014; Holt et al., 2012; Kay and Butenschön, 2018). However, these methods require a thorough representation of physical and biological processes with highly nonlinear dynamics. While they can simulate environmental parameters and their distribution mechanisms, they may not always achieve the accuracy needed for specific

applications (Storto et al., 2019; Tian et al., 2022).

  In contrast, synergizing the extensive coverage of satellite data with the high precision of in situ data represents an effective approach, enabling the frequent characterization of the ocean's vertical structure across an expanded spatial scope (Buongiorno Nardelli, 2020; Tian et al., 2022; Gao et al., 2024; Zhou and Zhang, 2023). Empirical models were widely used to extrapolate important ocean variables from the surface to deeper layers (Morel and Berthon, 1989; Uitz et al., 2006), but they

were vulnerable to inaccurate estimates due to the intricacy and non-linearity, particularly in locations with irregular vertical stratification and small-scale phenomena (Sammartino et al., 2020). Recent advancements in neural network (NN) technology have yielded promising results in addressing this issue (Asdar et al., 2024). For instance, Richardson et al. (2002) pioneered the use of an unsupervised NN for vertical chlorophyll reconstruction. Supervised NNs are capable of fitting nonlinear relationships between sea surface environmental variables and deep-sea conditions and have been successfully applied to the

estimation or prediction of various subsurface ocean parameters such as temperature and salinity (Buongiorno Nardelli, 2020; Qi et al., 2022; Smith et al., 2023; Su et al., 2021), and density (Su et al., 2024). Additional studies have supplemented sea surface parameters with reanalysis or profile data to reconstruct more subsurface parameters (Hu et al., 2023; Tian et al., 2022; Zhou and Zhang, 2023). However, due to the complex mechanisms and heterogeneous distribution of nitrate (Webb, 2021), its 3D reconstruction was not developed as effectively as parameters like temperature, particularly as the need for concurrent

vertical observations of additional variables persists. Wang et al. (2023) employed a regionalized deep neural network (DNN) to estimate nitrate concentration in the northwestern Pacific Ocean. Similar supervised techniques based on the Multilayer Perceptron (MLP) have been utilized to rebuild water-column bio-optical and biogeochemical variables using remote sensing and BCG-Argo data (Fourrier et al., 2020; Sauzède et al., 2017). A Bayesian strategy was proposed to supplement in situ data by inferring vertical profiles of unmeasured variables (Bittig et al., 2018). Yang et al. (2024) successfully reconstructed the 3D

nitrate structure of the Indian Ocean from surface data using two advanced artificial intelligence networks. However, this study relied on simulated data for supervised training instead of actual observational data, which may limit the model's applicability in real ocean environments.

  In this study, we develop an MLP to accurately reconstruct the 3D nitrate concentration upper 2000 m ocean, addressing the aforementioned challenges. Including vertical profile variables among the features might introduce potential uncertainty and

limit the expansion of the estimation range, so input features are exclusively based on sea surface environmental variables. The model employs a continual learning strategy (Kirkpatrick et al., 2017), initially pre-training on simulated-nitrate data to boost its generalization capabilities. The 3D nitrate field of the pan-European ocean from 2010 to 2023 is reconstructed based on this model and reveals the spatiotemporal distribution and interannual variations. Additionally, the contribution of each feature to the model estimates is calculated using Shapley values (Lundberg and Lee, 2017; Shapley, 1988), quantifying the effectiveness

of features.



**Figure 1.** The pan-European domain, including the MED and the NEA. The study area is highlighted in blue, with shades of color indicating ocean depth. The warm-colored grid indicates the count of BGC-Argo observations. Two rectangular boxes are selected as typical data regions for pattern comparison.

## 2 Material and methods

### 2.1 Study area

The study area extends from 30° W to 37° E latitude and 30° N to 65° N longitude, covering the Mediterranean Sea (MED) and a portion of the Northeast Atlantic (NEA). This area is considered to be coastal of the Pan-European domain, as shown in Figure 1. The study area focuses on the shelf seas around Europe, which play a disproportionately large role in the marine environment. Shelf seas contribute to 30% of marine primary productivity (Longhurst et al., 1995; Smith and Hollibaugh, 1993) despite covering less than 10% of the global seas (Holt et al., 2009). It is thus essential to accurately estimate important parameters of marine biogeochemical processes in these areas for understanding marine systems.


Nutrients from the open ocean and river runoff create a general and rapid biogeochemical cycle in the NEA (Gattuso et al.,
1998), which contrasts with the MED. The study aims to validate its effectiveness in different marine regions by exploring the
relationships among multiple variables.

## 2.2 Data

### 2.2.1 In situ nitrate data

The in situ data of nitrate concentration used in this study were obtained through BGC-Argo (https://argo.ucsd.edu, https:
//www.ocean-ops.org), a network of profiling floats equipped with sensors capable of monitoring six biogeochemical variables
(Claustre et al., 2020). The time, longitude, latitude, and pressure representing the depth are also recorded during the obser-
vations. Nitrate concentration is measured using ultraviolet absorption spectroscopy (Johnson et al., 2024), with an average
accuracy of $\pm 0.5 \mu mol \cdot kg^{-1}$ (Johnson et al., 2021, 2017; Mignot et al., 2019). In this study, a total of 477,870 data in the
study area are used, with 409,011 collected from the MED and 68,859 from the NEA.

The GLODAPv2 Database (https://doi.org/10.25921/1f4w-0t92) provides a uniformly calibrated open ocean data product
on inorganic carbon and carbon-relevant variables (Lauvset et al., 2022, 2021; Olsen et al., 2020). GLODAPv2 contains 15
cruises within the study area, utilized for independent validation of the model's predictive performance.

### 2.2.2 Matching sea surface environmental variables datasets

Sea surface environmental variables (SSEV) matched to in situ nitrate data are used as input features for the model, detailed in
Table 1. The SSEV data all span from 2010 to 2023, matching the BGC-Argo data since 2012 and enabling the reconstruction
of the 3D nitrate field since 2010.

The satellite-derived ocean color data were obtained from the European Space Agency's Global Color Project (Lavender
et al., 2009; Stéphane et al., 2010), with a spatial resolution of 25 km and a temporal resolution of monthly (http://globcolour.
info). The meteorological driver data were taken from the ERA5 reanalysis dataset (Hersbach et al., 2020) (https://cds.climate.
copernicus.eu), with a spatial resolution of 0.25° and the temporal resolution of monthly averaged reanalysis. ERA5 is the fifth
generation ECMWF atmospheric reanalysis of the global climate. Reanalysis combines model data with observations into a
globally complete and consistent dataset. The Copernicus Marine Service (CMEMS, https://marine.copernicus.eu/) provides
ocean dynamics-related data, which has a spatial resolution of 0.25° and a temporal resolution of monthly averages.

## 2.3 Methods

Figure 2 depicts the process of estimating nitrate and related research in this paper. The SSEVs and spatiotemporal coordinates
undergo data preprocessing and resampling (Section 2.3.1) to serve as the feature set for the two-step training of the MLP
model. The simulated and BGC-Argo nitrate concentrations provide the constructed MLP model (Section 2.3.2) with labels
for two-stage continual learning (Section 2.3.3) training. So far, MLP completes modeling the relationship between the surface
environment and internal ocean nitrate. After undergoing four kinds of 3D performance validations, the model reconstructed the





| Parameter | Description | Unit | Spatial resolution | Temporal Resolution | Data source |
|---|---|---|---|---|---|
| Chl | Chlorophyll concentration | mg· m$^{-3}$ | 25km | monthly | Globcolour |
| SPM | Inorganic suspended particulate matter | g·m$^{-3}$ | | | |
| NFLH | Normalised fluorescence line height | mW·cm$^{-2}$· microm$^{-1}$· sr$^{-1}$ | | | |
| CF | Cloud fraction | % | | | |
| PAR | Photosynthetically available radiation | einstein·m$^{-2}$ ·day$^{-1}$ | | | |
| CDM | Coloured dissolved and detrital organic materials absorption coefficient at 443 nm | m$^{-1}$ | | | |
| ZHL | Heated layer depth | m | | | |
| ZEU | Depth of the bottom of the euphotic layer | m | | | |
| ZSD | Secchi disk depth | m | | | |
| SST | Sea surface temperature | K | 0.25° | monthly | ERA5 |
| SP | Surface pressure | Pa | | | |
| TP | Total precipitation | m | | | |
| Z | Total depth | m | | | |
| U10 | 10 m U wind component | m · s$^{-1}$ | | | |
| V10 | 10 m V wind component | m · s$^{-1}$ | | | |
| S10 | 10 m wind speed | m · s$^{-1}$ | | | |
| SSH | Sea surface height | m | 0.25° | monthly | CMEMS |
| MLD | Density ocean mixed layer thickness | m | | | |

**Table 1.** Details of the SSEV dataset.

3D nitrate field by inputting iterated spatiotemporal coordinates and corresponding SSEV datasets. The feature contributions and the potential mechanisms for estimation was evaluated based on the training datasets and the model (Section 2.3.5).

### 2.3.1 Data pre-processing

The candidate input variables for estimating nitrate are depth, latitude, longitude, day of the year, and SSEV data mentioned in Section 2.2.3. The time variables (day of the year) and geographical coordinates (latitude, longitude and depth) are intended

to explain the temporal and spatial variations of the studied parameters. The characteristics of biogeochemistry in the ocean properties are described by SSEV such as SST and Chl (D'Ortenzio and Ribera d'Alcalà, 2009). Furthermore, variables such as SSH provide insights into the ocean dynamics, which may contribute to obtaining more accurate vertical stratification.





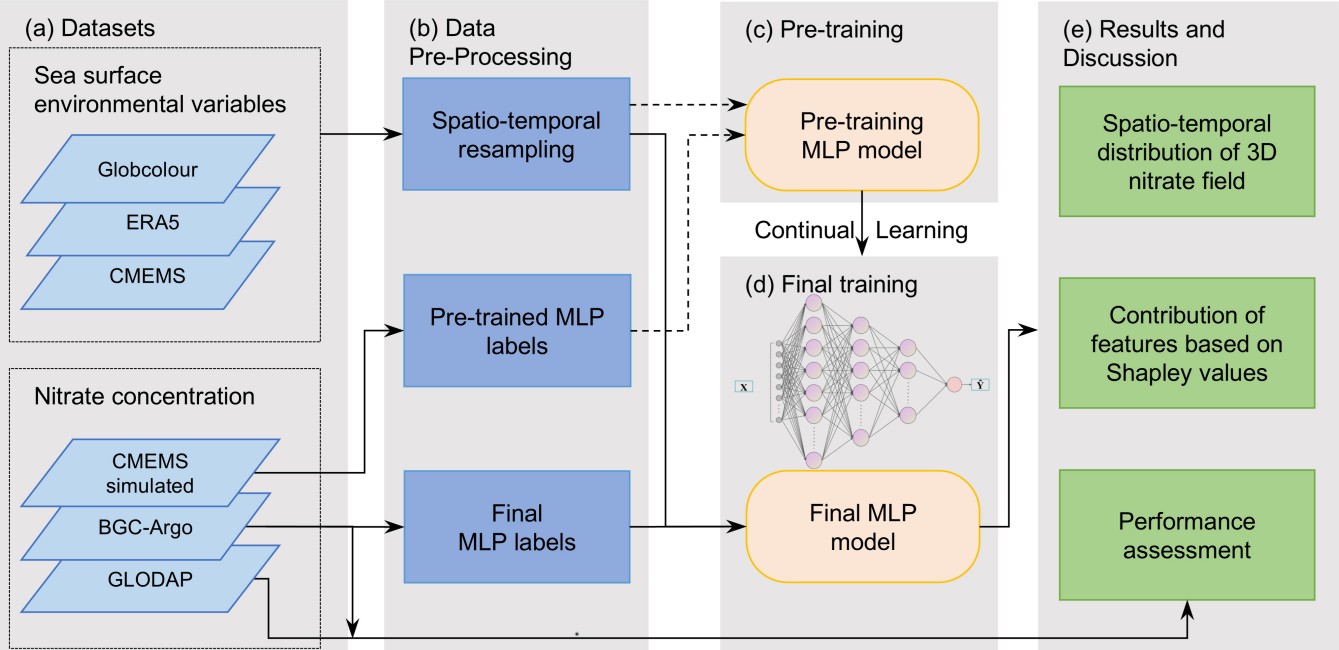

**Figure 2.** Workflow for nitrate estimation and research on reconstructed results.

All these predictor variables are utilized as input features after preprocessing to train the MLP, with nitrate concentrations serving as supervisory (Figure 2b). The potential uncertainty in the input features can be significantly reduced by implicitly

incorporating them into the weights of the model when utilizing the same data products (Chen et al., 2019). The gridded SSEV data are interpolated to obtain features with the spatiotemporal correspondence of the BGC-Argo data. For the SSEV missing data, estimates that could be obtained after limited interpolation are retained, while the training data with more severe missing values are excluded. To utilize the annual period, the sampling dates are projected onto the circular coordinates as follows:

$$\mathrm{Jday1} = \cos(2\pi \cdot (\mathrm{dayofyear}/365)), \qquad (1)$$


$$\mathrm{Jday2} = \sin(2\pi \cdot (\mathrm{dayofyear}/365)). \qquad (2)$$

The other input features are then normalized by applying Z-score transformations as follows:

$$z(x_i) = (x_i - \mu)/\sigma, \qquad (3)$$

where $\mu$ and $i$ are the mean and standard deviation of each feature of the train set, $x_i$ is the input value of feature $i$. Z-

score transformation is a linear normalization technique commonly used in MLP development to align the inputs and intended outputs within comparable value ranges.



### 2.3.2 Multilayer perceptron

The study develops a Multilayer perceptron (MLP) (Bishop, 1995) model, which is a type of feed-forward neural network that can be used for various types of input or output mappings (Hagan et al., 1997). MLPs can approximate any continuous and derivable function by means of an error backpropagation algorithm (Rumelhart et al., 1986). An MLP consists of interconnected neurons organized into input, hidden, and output layers. Each connection is assigned a weight '$w$', and the output is generated by combining inputs and weights using an activation function after adding the neuron's bias '$b_j$'. The weights are iteratively updated during the training epochs to minimize the loss function which reduces the quadratic error between MLP outputs and labels. This iterative process continues until a minimum is reached using the approach of error backpropagation.

The structure of the MLP is determined by a series of experiments with multiple hidden layers, and it utilizes the LeakyRelu activation function. The optimal network is determined through multiple trials, where the structure with the least amount of error on the test dataset and the fewest neurons is selected. The final network was configured as (22-128-64-16-1), comprising one input layer with 22 inputs, three hidden layers with 128, 64, and 16 nodes, and one output layer with the nitrate concentration as the output value.

### 2.3.3 Continual learning

The generalization capability of deep learning models, including MLPs, is highly dependent on the representativeness of the training data. Insufficient or imbalanced training data can exacerbate generalization errors and increase the risk of model overfitting. In the domain of water resource research, challenges associated with the collection of in situ data have highlighted the effectiveness of transfer learning (TL) techniques(Cao et al., 2020; Harkort and Duan, 2023; Miao et al., 2023; Syariz et al., 2020; Zhu et al., 2017). Nevertheless, most TL applications are based on fine-tuning (Ma et al., 2024), which limits their capacity to integrate knowledge from multiple datasets in a more comprehensive manner (Zhou and Zhang, 2023). To overcome this limitation, we developed the continual learning (CL) strategy to improve the training process of BGC-Argo. CL enables the model to assimilate new knowledge continually while retaining previously acquired information, thereby enhancing the robustness and adaptability of the model.

In practice, the simulated nitrate data are initially employed for pre-training, after which the derived network weights are transferred to the subsequent training phase supervised by BGC-Argo observations. Ideally, this sequential process enables the model to capture the general distribution patterns and underlying variation mechanisms present in the simulated nitrate data, and subsequently refine its estimations to achieve higher accuracy using BGC-Argo measurements. However, when the model undergoes incremental training through gradient-based updates, it may experience catastrophic interference or forgetting, leading to the degradation of previously acquired knowledge (Kirkpatrick et al., 2017). To address this issue, Elastic Weight Consolidation (EWC), a regularization-based continual learning algorithm, is applied to constrain weight updates by assigning greater importance to critical network parameters (Kirkpatrick et al., 2017).

Figure 3 illustrates the effect of training strategies on the two-stage training task and how EWC ensures the retention of knowledge from Task A during the learning of Task B. Sets A and B represent the solution spaces for two training tasks,





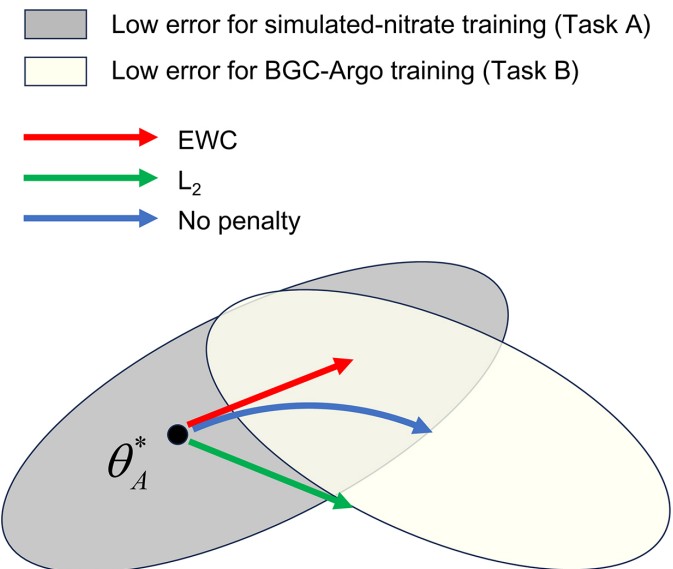

**Figure 3.** Schematic illustration of how training strategies influence study trajectories in a two-stage task.

specifically the simulated nitrate and BGC-Argo training. After completing the Task A, the parameters are denoted as $\theta_A^*$, and the three trajectory lines depict different training processes under varying loss function constraints. Constraining each weight equally (green arrow) imposes excessively rigid restrictions, allowing Task A to be retained only at the cost of failing to learn Task B. Conversely, applying gradient steps based solely on Task B (blue arrow) effectively minimizes the loss for Task B but compromises the knowledge acquired from Task A. Although BGC-Argo measurements are accurate, they are limited in their

spatiotemporal coverage for nitrate reconstruction studies, which is insufficient for a comprehensive global characterization of nitrate distributions. Consequently, the center of Set B represents the optimal solution for model weights on the BGC-Argo training set, but which is an overfitted and suboptimal for broader global reconstruction. In contrast, the EWC trajectory (red line) finds an optimal balance for Task B while calculating the importance of weights for Task A, thus ensuring minimal loss in Task A's performance. Robust weights should lie between Sets A and B, balancing the broad and generalizable knowledge

from simulated nitrate with the precise measurements from BGC-Argo. This process can be understood as guiding the model to retain the broad knowledge to enhance the generalization ability of Task B, or as calibrating the simulated nitrate with the precision of BGC-Argo. Given that simulated nitrate provides concentration data across the entire ocean, especially in regions not yet observed by BGC-Argo, this strategy is crucial for enhancing the generalization capability and robustness of the MLP model.

EWC relies on the Fisher Information Matrix (FIM) to estimate the importance of each model parameter concerning previous tasks (Fisher and Russell, 1997). The FIM quantifies the amount of information that an observable random variable carries about an unknown parameter, reflecting how sensitive the likelihood function is to changes in the parameters. The FIM $F$ is





defined as:

$$F = \mathbb{E}_{x,y\sim p_{\text{data}}(x,y)}\left[\nabla_\theta \log p(y|x;\theta)\nabla_\theta \log p(y|x;\theta)^\top\right],\tag{4}$$

where $p(y|x;\theta)$ is the likelihood of the target $y$ of the given data $x$ and model parameters $\theta$, and $\nabla_\theta \log p(y|x;\theta)$ is the gradient of the log-likelihood with respect to the parameters.

In practical applications, computing the full FIM is computationally expensive, particularly for large neural networks. To simplify the computation, it is often assumed that the FIM is diagonal, effectively ignoring dependencies between parameters. In MLPs, the diagonal elements of the FIM can be approximated as follows:

$$F_i \approx \mathbb{E}_{x,y\sim p_{\text{date}}(x,y)}\left[\left(\frac{\partial \log p(y|x,\theta)}{\partial \theta_i}\right)^2\right].\tag{5}$$

Since the true distribution of data is unavailable, training data is typically used for estimation:

$$F_i \approx \frac{1}{N}\sum_{n=1}^N \left(\frac{\partial \log p(y_n|x_n,\theta)}{\partial \theta_i}\right)^2,\tag{6}$$

where, $N$ is the number of training samples, $(x_n, y_n)$ are the data samples, and $\theta_i$ is the $i$-th parameter of the model.

In regression tasks using MLPs, we model the output $y$ as:

$$y = f(x;\theta) + \epsilon, \quad \epsilon \sim \mathcal{N}(0,\sigma^2).\tag{7}$$

Assuming Gaussian noise with constant variance $\sigma^2$, the diagonal elements of the FIM can be approximated based on the gradients of the model's output with respect to its parameters. Specifically, we compute $F_i$ as:

$$F_i \approx \frac{1}{N}\sum_{n=1}^N \left(\frac{\partial f(x_n;\theta)}{\partial \theta_i}\right)^2,\tag{8}$$

where $x_n$ is the $n$-th input sample, and $\frac{\partial f(x_n;\theta)}{\partial \theta_i}$ is the partial derivative of the model output with respect to parameter $\theta_i$. This

approximation allows efficient computation of $F_i$ during training.

In the Bayesian framework, the goal is to find the parameter $\theta$ that maximizes the posterior probability given both the previous task data $D_A$ and the new task data $D_B$:

$$p(\theta|D_A, D_B) \propto p(D_B|\theta)p(\theta|D_A).\tag{9}$$

Since directly computing $p(\theta|D_A)$ is intractable, we approximate it using a Gaussian distribution centered at the previous

optimal parameters $\theta_A^*$ with precision given by the FIM (MacKay, 1992):

$$p(\theta|D_A) \approx \mathcal{N}(\theta_A^*, F^{-1}).\tag{10}$$

Taking the negative logarithm of the posterior and ignoring constants independent of $\theta$, we obtain the total loss function:

$$L_{EWC}(\theta) = L_B(\theta) + \frac{\lambda}{2}\sum_i F_i(\theta_i - \theta_{A,i}^*)^2,\tag{11}$$





where $L_B(\theta)$ is the loss for the new task only, $i$ labels each parameter, $F_i$ is the FIM of the previous task, $\theta_A^*$ is the optimal

parameter value after training on the previous task, and $\lambda$ is a hyperparameter controlling the trade-off between performance

on the new task and retention of the previous task knowledge.

### 2.3.4   Model validation

Nitrate concentrations derived from the identical vertical observations by BGC-Argo exhibit a strong correlation and a gradual

variation with increasing depth. Conventional methods that divide the entire dataset proportionally can result in highly similar

data appearing in both the training and test sets, thereby leading to an exaggerated model performance (Salazar et al., 2022).

Hence, it is imperative to partition the BGC-Argo dataset based on the observation period, with each period referred to as a

profile (Sammartino et al., 2020; Sauzède et al., 2017). This division method ensures the identical distribution and independence

of the training and testing sets. Furthermore, the spatial generalization capabilities of the model can be further assessed by

partitioning the dataset based on devices.

A 5-fold cross-validation approach is employed to evaluate the model performance using independent test data, which is

widely used in statistics and machine learning. In each cross-validation, the entire observational dataset is divided into a

training set (80%) and a testing set (20%) based on profiles. The training set is utilized to train the MLP model, from which the

reconstruction is subsequently derived. The testing set is employed as an independent dataset to evaluate the performance of

the model. On the test set, the MLP performance was evaluated by comparing the estimated values with in situ nitrate values,

using statistical metrics including the determination coefficient ($R^2$), mean bias error (MBE), mean square error (MSE), root

mean square error (RMSE), mean absolute error (MAE), and median absolute error (MedAE).

### 2.3.5   Evaluating the contribution of inputs

Another major limitation of MLPs and deep learning networks is the lack of interpretability, which makes it challenging to

evaluate the estimation processes and mechanisms. However, it is essential to assess the validity of environmental parameters

for estimating nitrate concentration, especially since their influence and interactions are not fully elucidated.

The Shapley values (Shapley, 1988) is a method in coalition game theory that effectively describes how benefits are fairly

distributed among contributions by the difference between the predicted and average predicted values in each case. The Shapley

value of a feature is its weighted and summed contribution to the output over all possible feature combinations:

$$\phi_j(val) = \sum_{S \subseteq \{1,\ldots,p\} \setminus \{j\}} \frac{|S|!(p-|S|-1)!}{p!} (val(S \cup \{j\}) - val\{S\}), \tag{12}$$

where $\phi_j$ is the contribution of the j-th feature to the results. $S$ is a subset of the model's features, and $p$ is the total number of

features. $val(S)$ is the prediction for feature values in S that are marginalized over features not included in $S$.

Shapley additive explanations (SHAP) (Lundberg and Lee, 2017) is a method for explaining individual estimation results

based on Shapley values, which has been successfully applied to evaluate predictors using machine learning algorithms in

environmental research (Hu et al., 2023). The purpose of SHAP is to compute the contribution of each feature to the result



Earth System
Science
Data

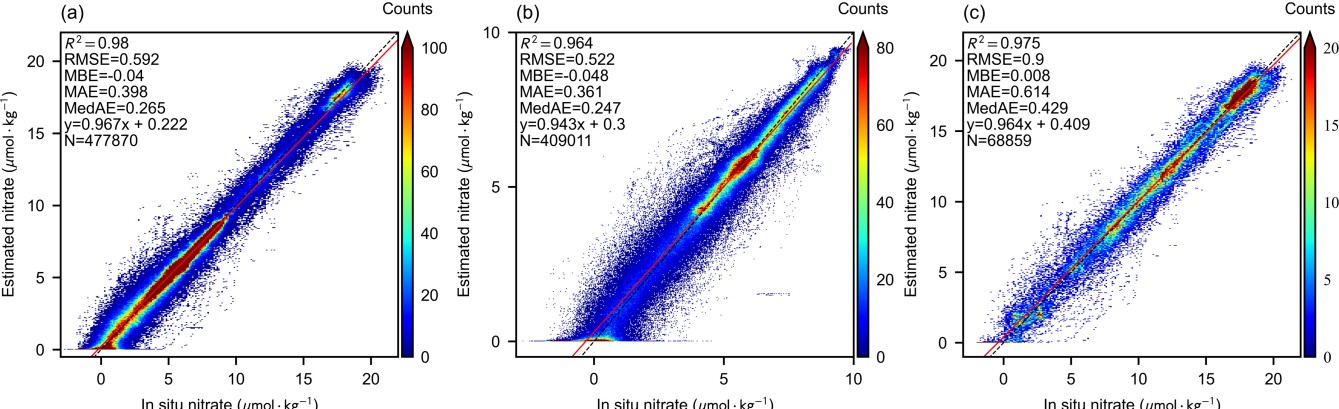

**Figure 4.** Estimation performance on the test set validated by BGC-Argo measurements (a). The test set results are further divided into the MED (b) and NEA (c) regions. The red line indicates the fitted trend of the data while the black dashed line denotes the 1:1 parity line.

to explain an instance. The Shapley values are depicted as a linear additive feature attribution approach. SHAP specifies the explanation as:

$$g\left(x'\right) = \phi_0 + \sum_{j=1}^{M} \phi_j, \tag{13}$$

where $g$ is the model to be explained, $M$ is the size of feature space and $\phi_j \in R$ is the feature contribution for feature $j$, as same as the Shapley values of $j$.

We can calculate SHAP to quantify the contribution of each feature to the prediction results of a black box model in different samples. The feature tends to increase the output result when SHAP is positive. Conversely, the feature tends to decrease the output result when SHAP is negative. Absolute SHAP value (ASV) indicates the degree that the feature affects the output. To observe the overall significance, the mean of ASV for each feature in the data is therefore defined as:

$$I_j = \frac{1}{n} \sum_{i=1}^{n} \left| \phi_j^{(i)} \right|, \tag{14}$$

where $i$ represents data samples, and $j$ represents features.

## 3 Results and discussion

### 3.1 Model performance

In the 5-fold profile-based cross-validation, all data is used once in the test set. As illustrated in Figure 4, the model performance is evaluated by comparing estimated values with BGC-Argo observations. the model demonstrates high accuracy in estimating

nitrate concentration, with estimated values generally aligning along the 1:1 line. Importantly, to ensure a comprehensive





dataset and enhance the stability of the reconstruction process, we retained all measured labels, including negative values. Furthermore, a Softplus activation function was applied to the model's output layer to guarantee non-negative predictions, albeit at the expense of some degradation in statistical performance metrics. Considering the significant differences between the two regions of the study area, Figure 4b&c respectively show the test results for the MED and NEA. Compared to the NEA,

the MED exhibits a smaller range of nitrate variations and demonstrates stronger estimation performance. The MED records account for 86% of the total dataset, whereas the NEA contributes only 14%. This data imbalance likely contributes to the more consistent performance in the MED compared to the NEA.

While the model has shown satisfactory overall performance, it is critical that the accuracy remains consistently desirable in the vertical dimension. Only then can the model fulfill its intended purpose of estimating and reconstructing the entire 3D

ocean nitrate field. Figure 5 illustrates the vertical distribution of the primary statistical metrics and their comparison with simulated-nitrate. The model maintains robust performance in the vertical dimension, with no significant fluctuations in RMSE and MBE, which is vital for accurately estimating nitrate profiles. The model exhibits slightly superior performance in the MED compared to the NEA at most depths. The RMSE in both the MED and NEA is higher between 0 m and 150 m, with a notable peak at 60 m depth, reaching about 0.8 and 1.4 $\mu mol \cdot kg^{-1}$ respectively. Furthermore, the RMSE of MED remains at

a low level and slowly decreases as depth increases. In contrast, the RMSE of the NEA varies drastically, accompanied by a larger overall error, particularly in the 400-700 m depth range. As shown in Fig. 5b, MBE values are negative for most depth ranges in NEA, suggesting a slight overall underestimation of nitrate concentrations, while a slight overestimation occurs in the upper ocean layers of both sub-regions

The model excels in the deep ocean layers beyond 800 m depth, where nitrates are characterized by low variability and

minimal feedback from the sea surface environment. Nonetheless, through the use of temporal and spatial coordinates and a dual training process, the model accurately estimates nitrate concentrations. The model's relatively weak performance is observed in the upper 100 m depths, which could be attributed to the sensitivity of the surface layer to external nitrate inputs (Altieri et al., 2021), thus leading to deviations from the model-fitted relationship between nitrate and SSEV. Furthermore, the ocean at these depths is usually influenced by both the euphotic layer and mixed layers, where complex interactions between

ecosystem and ocean dynamics occur, such as water transport, plankton consumption and decomposition. Hence, predicting parameters within this depth has usually presented the biggest challenge in vertical dimension estimation (Sammartino et al., 2020).

In contrast, simulated-nitrate exhibits instability in describing the vertical distribution of nitrate concentration. Firstly, simulated-nitrate produces significant errors at the ocean surface, possibly due to the limitations of biogeochemical models

in simulating complex boundary interactions. However, the estimation error here has been significantly ameliorated by MLP owing to the strong correlation between SSEV and SSN. Secondly, the characterization of nitrate vertical changes by simulated-nitrate is not precise enough. The vertical inaccuracy may result in disparities in the characterization of the changes, thus posing a limitation for small-scale biogeochemical research. An evident instance can be observed in the mesopelagic zone (MZ) layer, ranging from 200 to 1000 m in the NEA. Simulated-nitrate faced challenges in accurately describing the vertical rate of nitrate

variations in this range, resulting in a notable overestimation of nitrate concentrations (Fig. 5b). This is also reflected in the

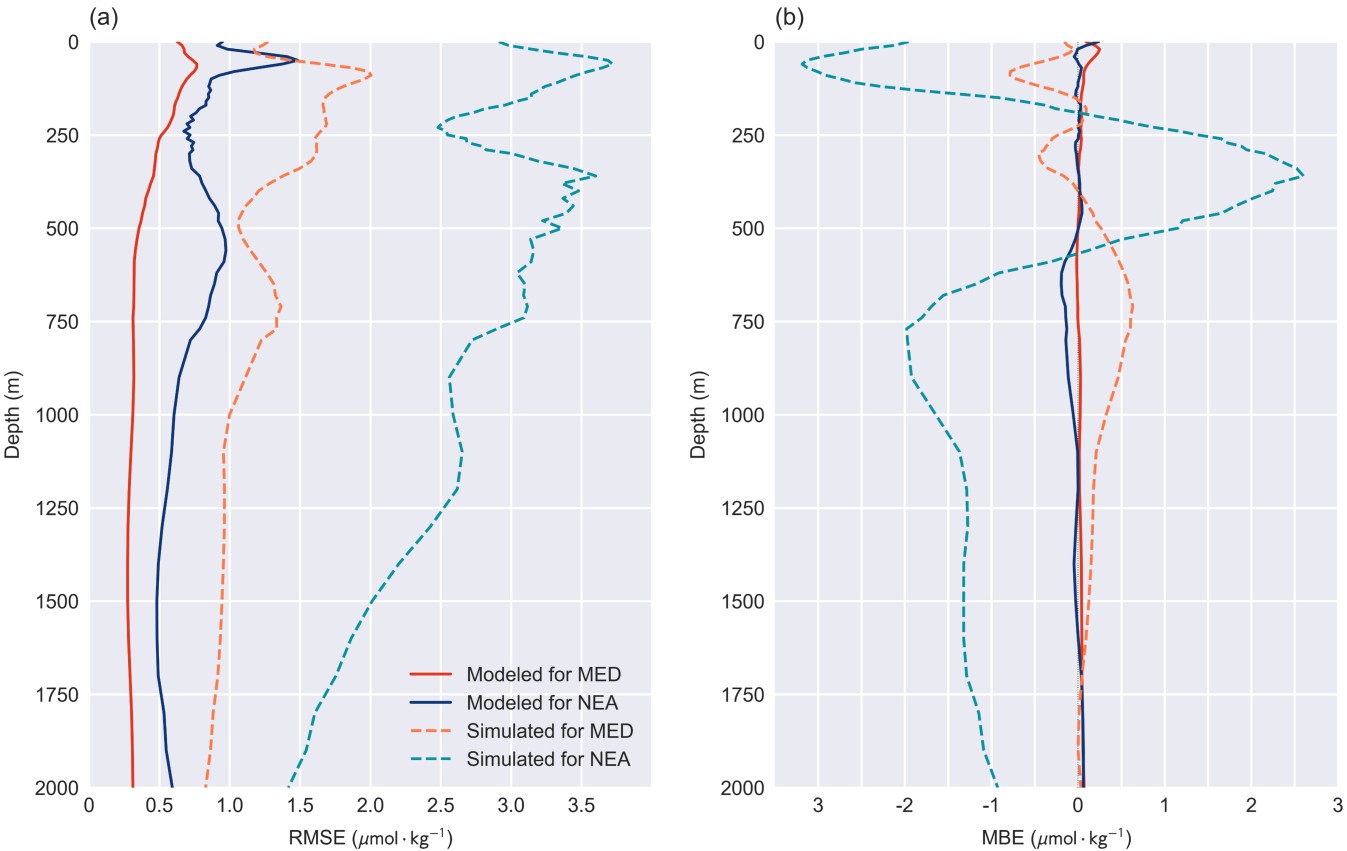

**Figure 5.** Vertical profiles of RMSE (a) and MBE (b) for modeled and simulated nitrate compared to BGC-Argo measurements.

distinct step-like pattern observed at nitrate concentrations of 10-15 $\mu mol \cdot kg^{-1}$ (Fig. 6a) and the overestimation in the vertical pattern (Fig. 8e). Furthermore, MLP and simulated nitrate present similarities in the vertical profiles of RMSE, while MLP consistently outperforms simulated-nitrate. This also demonstrates that MLP has improved performance by incorporating prior knowledge from simulated-nitrate through CL.

## 3.2 Spatial generalization ability and enhancement of continual learning

Due to the limited monitoring range of BGC-Argo, the spatial generalization ability of the model is crucial for accurately reconstructing the complete nitrate concentration field. In this section, a more rigorous cross-validation is implemented, which divides the dataset based on the BGC-Argo device, as shown in Fig. 6. This division approach facilitates the evaluation of the predictive performance in unobserved ocean regions and enables a further comparison of the effect of CL on the generalization ability.

Figure 6e demonstrates the accuracy of the simulated-nitrate, which is compared by interpolating it along temporal and spatial dimensions to match the BGC-Argo observation coordinates. The results indicate a general consistency between them, de-





spite certain discrepancies in capturing local variability, resulting in step-like cluster formations. The simulated-nitrate achieves an acceptable accuracy ($R^2$ = 0.826, RMSE = 1.882 $\mu mol \cdot kg^{-1}$), making it a valuable prior dataset. This compatibility is
crucial, and given that simulated-nitrate can provide data of comparable accuracy across the entire ocean, its shows great potential as a complement to BGC-Argo data. Simulated nitrate aids in understanding the large-scale distribution of nitrate, offering extensive insights that serves as a beneficial foundation for enhancing fitting relationships during subsequent CL training phases.

Figure 6a illustrates the spatial generalization test performance of the model, demonstrating that CL leads to enhanced test
performance, marked by an increase in $R^2$ of 0.051 and a decrease in RMSE of 0.343 $\mu mol \cdot kg^{-1}$, compared to the non-CL results shown in Fig. 6c. Specifically, the MLP model without CL significantly underestimates high nitrate concentration samples, primarily due to its limited generalization ability in unfamiliar regions of the NEA during site-specific cross-validation. The introduction of CL effectively mitigates this limitation, allowing the model to maintain stable generalized estimates, even for high nitrate concentration samples. Furthermore, the majority of samples exhibit a more consistent fit to the 1:1 line, sig-
nificantly reducing episodic uncertainty associated with simulated nitrate and the generalization error of the non-CL MLP model. Notably, coupling with CL retains the influence of prior knowledge from simulated nitrate, resulting in localized variability differences. For instance, the densely packed high-concentration samples in warm colors transitioned from symmetric fitting (Fig. 6c) to step-like clustering (Fig. 6a), but achieving a closer fit and overall improved performance. The extent of this transformation is influenced by the EWC parameter $\lambda$.

Figure 6b presents the horizontal distribution of RMSE. The predictions exhibit the highest accuracy in the central MED, while larger RMSE values are observed in the NEA and peripheral regions of the MED. The substantial variability in nitrate concentrations in the NEA is largely attributable to the active exchange of eutrophic seawater. Furthermore, the sparse distribution of BGC-Argo measurements in the NEA results in significant deviations from the training data, posing substantial challenges for accurate cross-validation in this region. Overall, high error rates are frequently observed in coastal locations,
particularly in the western Strait of Gibraltar and southern parts of the MED. These areas are more susceptible to anthropogenic influences and complex land-sea interactions, which complicate prediction efforts. Additionally, the shallower topography of these regions contributes to increased errors in the vertical water column, particularly in the error-prone ocean surface layer (Fig. 5a).

Notably, regional disparities introduced by CL are evident in Fig. 6d. When the peripheral regions are used as test sets, the
discrepancies between training and test data distributions become more pronounced. This sparsity of BGC-Argo data poses a considerable challenge for model estimation in regions lacking sufficient global training on similar datasets, leading to reduced performance metrics. However, CL significantly enhances the model's estimation capabilities in sparsely observed regions, particularly in areas with high RMSE near the boundaries of BGC-Argo coverage. This suggests that CL helps reduce model instability when generalizing to unfamiliar regions by incorporating prior knowledge from simulated nitrate. For instance,
in the western Strait of Gibraltar, complex environmental interactions and similarities in spatial coordinates with the MED present significant estimation challenges. Nevertheless, the model demonstrates substantial improvements in both accuracy and generalization stability compared to the MLP without CL. Moreover, the model achieves more accurate estimates by fitting

**Figure 6.** Performance of the model in site-based cross-validation on the test set(a), spatial distribution (b), and model comparison (c-f). (c) shows the test performance without using continual learning, along with the spatial distribution of increased RMSE (d). (e) depicts the validation performance for simulated nitrate concentrations and the spatial distribution of increased RMSE compared to this model.

Earth System

Science

Data

BGC-Argo data, showing a comprehensive improvement over simulated nitrate (Fig. 6f), which is critical for reconstructing the three-dimensional nitrate field. Interestingly, in data-dense regions such as parts of the Mediterranean, the incorporation of CL results in a slight increase in RMSE. This phenomenon occurs because, in well-sampled regions, prior knowledge may interfere with the MLP's fitting process. However, the influence of this prior knowledge can be optimized by regionally adjusting the EWC parameter $\lambda$. Currently, this parameter has been selected to achieve an overall optimal performance.

In conclusion, it is reasonable to infer that CL enhances the overall model performance and generalization capability in regions not covered by BGC-Argo by incorporating relevant knowledge and patterns from simulated nitrate. This process is influenced by data distribution and weighting parameters.

### 3.3 Independent validation with GLODAPv2

To avoid potential autocorrelation within the BGC-Argo dataset, the GLODAPv2 dataset was employed for independent validation. Fifteen cruises measuring nitrate concentrations at depths ranging from 0 to 2000 m in the study area were selected, and their measurements were compared against model estimates and simulated nitrate. Figure 7c illustrates that a significant portion of the GLODAPv2 data is located in the NEA, thereby allowing for an effective assessment of the model's performance in under-sampled regions. The results indicate a strong correlation with the GLODAPv2 nitrate concentrations, as evidenced by an $R^2$ value of 0.94 (Fig. 7a).

Figure 7c also depicts the regional distribution of errors, revealing that model performance varies significantly across different regions. The lowest errors are observed in the western MED and the southern NEA, whereas larger errors are concentrated in areas with sparse BGC-Argo observations. This distribution pattern and its underlying causes align with the spatial performance in site-based cross-validation, as shown in Fig. 5b. Factors such as enhanced water exchange dynamics (Berglund et al., 2023), intricate land-sea interactions, and shallow topography contribute additional complexities. Moreover, several expeditions north of 50°N occurred in 2010 - 2012 and 2015, while most BGC-Argo observations in the same region were conducted post-2020. The pronounced variability of nitrate concentrations in the NEA, coupled with limited observations and temporal discrepancies, diminishes the data representativeness, leading to increased RMSE. Overall, the error margins are deemed acceptable and still outperform the validation accuracy of simulated nitrate (Fig. 7b).

### 3.4 Validation of three-dimensional nitrate pattern

To thoroughly examine and contrast the seasonal vertical patterns of BGC-Argo, modeled, and simulated nitrate, two representative regions depicted in Fig. 1 were selected to facilitate a comprehensive evaluation of the model's efficacy. The Box NEA, situated at 14-19° W and 46-53° N, and the Box MED, located at 26-30° E and 32-36° N, were chosen due to their frequent BGC-Argo sampling, which ensures high consistency between measured data and observed vertical patterns. The vertical distribution of nitrate is depicted in Fig. 8 for these two regions, where the MLP model, simulated nitrate, and BGC-Argo observations are juxtaposed for comparison.

The reconstructed vertical nitrate profiles derived from the MLP model demonstrate greater consistency and robustness compared to BGC-Argo data, whereas the profiles from simulated nitrate still exhibit significant discrepancies in capturing

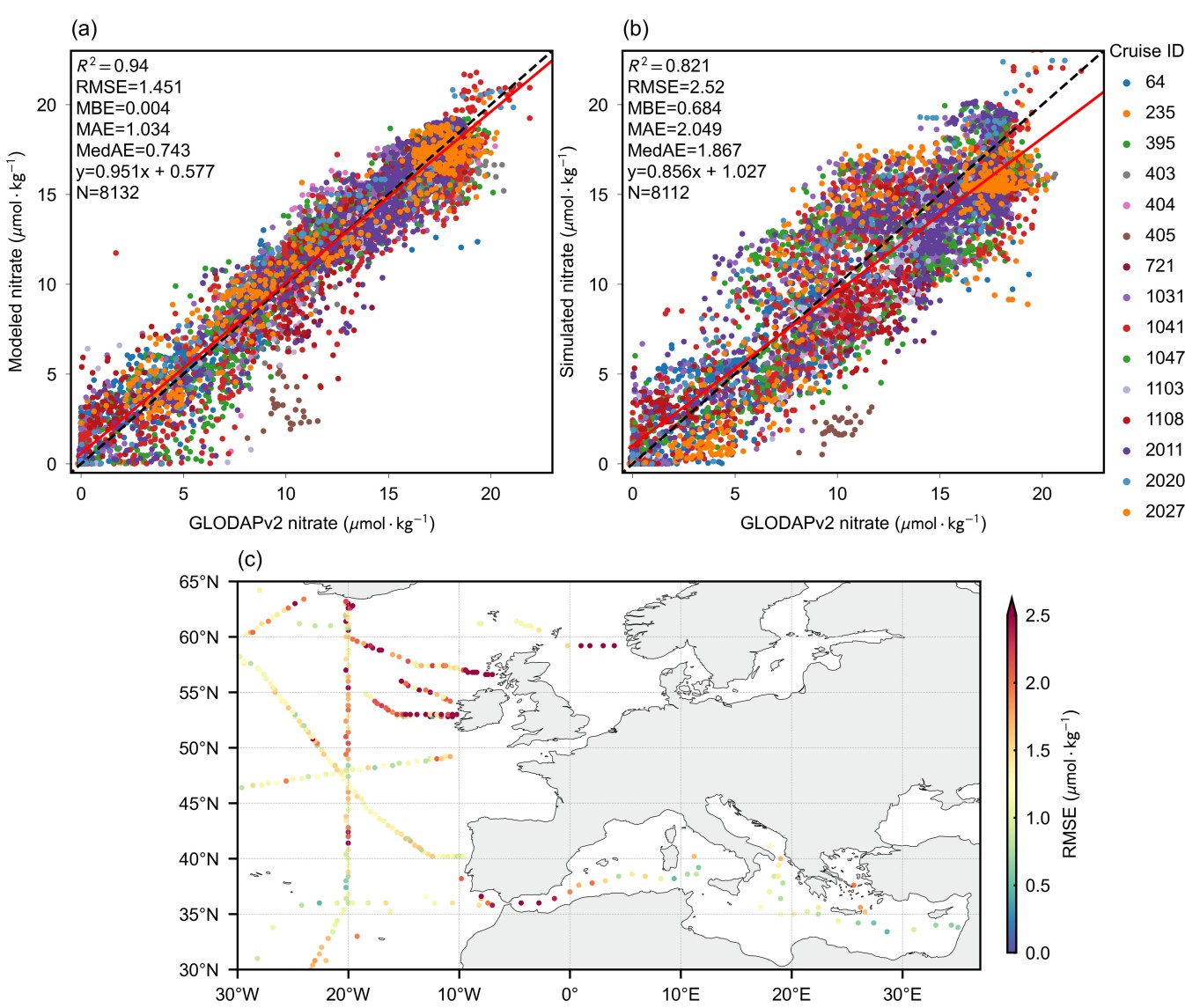

**Figure 7.** Accuracy comparison between MLP-estimated nitrate (a) and simulated nitrate (b) validated by GLODAPv2 measurements, with colors representing different cruise numbers. Figure (c) shows the spatial RMSE distribution map, where the RMSE at each point is calculated from the average vertical measurement error for each cruise location.





seasonal variations. The MLP model has shown a remarkable ability to capture mid-scale features, such as the seasonal increase during winter and decrease during summer, aligning well with BGC-Argo data and effectively depicting seasonal variations in the upper ocean. Furthermore, the MLP model is consistent with BGC-Argo in representing depth-dependent variability within the Mesopelagic Zone (MZ), while bridging gaps left by the intermittent nature of BGC-Argo measurements. In contrast,
simulated nitrate tends to underestimate concentrations in the upper ocean (Fig. 5b) and shows relatively sluggish seasonal variations, including the absence of a pronounced increase during winter in Box NEA and insufficient representation of seasonal changes in the upper layers of Box MED.

As previously discussed, the model's estimation results demonstrate satisfactory accuracy and strong performance on the test dataset. Additionally, the model's predictions have been analyzed comprehensively across vertical, horizontal, and temporal
dimensions, all indicating high performance. To ensure robust generalization across diverse oceanic environments, the joint model was employed to estimate nitrate concentrations in both the MED and NEA, despite facing specific challenges. Given the complexity of the marine environment and the fact that the NEA represents only 14% of the dataset, a higher error rate in this region is acceptable. Despite increased errors in some challenging cases, the model generally proves to be a reliable approach for reconstructing the 3D nitrate concentration field.

## 3.5 Spatial and temporal distribution of the reconstructed nitrate field

The reconstruction of the 3D nitrate field from 2010 to 2023 was conducted using the MLP model combined with the SSEV for the corresponding period. The reconstructed field features a monthly temporal resolution, a horizontal spatial resolution of 0.25 degrees, and 63 depth levels, with vertical intervals ranging from 5 to 50 m. Figure 9 illustrates the spatial distributions of the reconstructed nitrate at various depths across the pan-European region, with representative profiles selected at depths of 0,
50, 100, 150, and 500 m.

The reconstructed nitrate field reveals substantial spatial variability, with a clear increasing trend in nitrate concentrations with depth. The MED identified as an oligotrophic region, generally exhibits nitrate concentrations below 5 $\mu mol \cdot kg^{-1}$ at depths between 0 and 150 m. Unlike the NEA, the MED nitrate concentrations are less influenced by seasonal dynamics, primarily due to the region's enclosed nature and restricted seawater exchange. The oligotrophic characteristics of the MED
intensify from west to east, with more pronounced differences evident in the deeper ocean layers (Pujo-Pay et al., 2011; Ribera d'Alcalà et al., 2003). In contrast, the NEA is characterized by higher nitrate concentrations and pronounced seasonal variability, which is largely driven by the influx of nutrient-rich water masses from the open ocean (Berglund et al., 2023). The highest nitrate concentrations are found in the northern NEA seawaters, a typical eutrophic region.

Figure 10 presents the temporal-depth profiles of nitrate concentration as a function of the month in both the MED and
NEA, allowing for a more detailed examination of their temporal patterns. The seasonal variability of nitrate in the MED is relatively subtle. During winter, upwelling of nutrient-rich cold water elevates nitrate concentrations in the upper ocean, with a marked increase observed from October to February of the following year. After reaching this peak, nitrate levels decline due to phytoplankton uptake during spring, followed by a secondary rise in autumn as phytoplankton biomass decays (Severin et al., 2017). Conversely, the NEA exhibits a pronounced temporal pattern in nitrate concentrations, primarily governed by ocean





**Figure 8.** Comparison of monthly vertical patterns of nitrate in the designated region among BGC-Argo, the MLP model, and simulated climatology. (a), (c), and (e) correspond to the box MED, while (b), (d), and (f) correspond to the box NEA. The black dashed lines represent the average MLD from the CMEMS dataset.



**Figure 9.** Spatial distribution of reconstructed nitrate field, with columns representing four seasons and rows representing five depth slices.



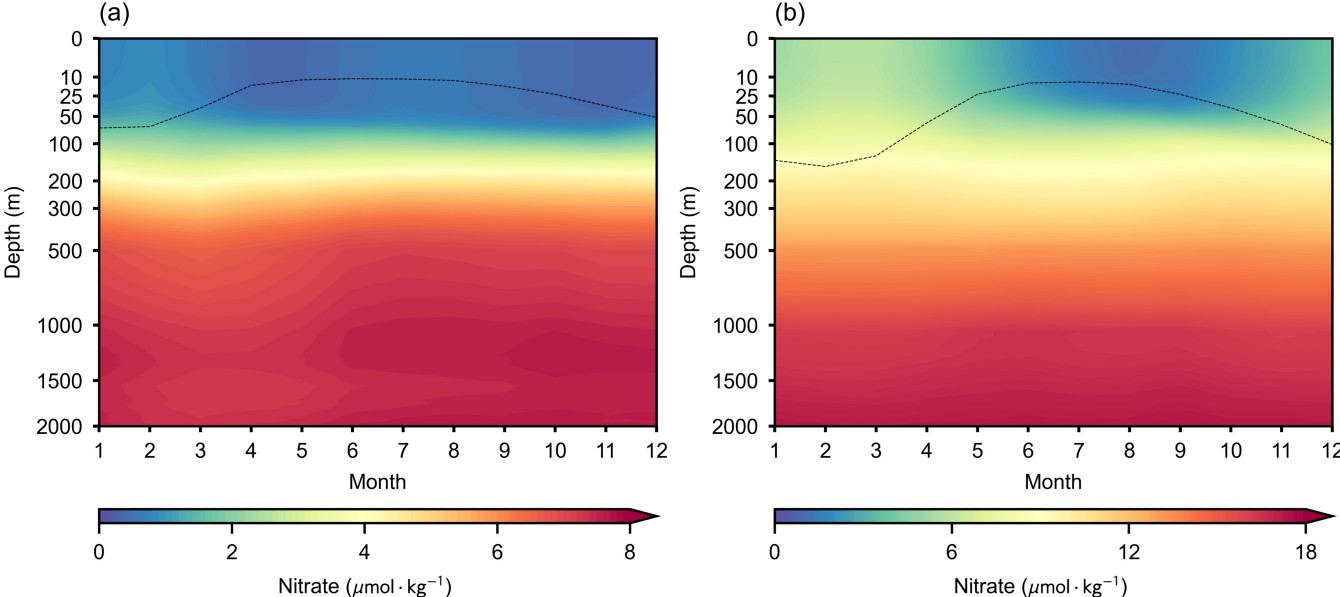

**Figure 10.** Monthly distribution profiles of nitrate concentrations in the MED (a) and NEA (b). The black dashed lines represent the average MLD from the CMEMS dataset.

dynamics. In the mixed layer, from the surface to approximately 100 m depth, nitrate levels increase from October to March and subsequently decrease from April to August. The temporal pattern in the NEA resembles the winter increase observed in the MED but lacks a distinct secondary peak, instead showing a more sustained high-nutrient period.

Figure 11 illustrates the interannual anomalies of nitrate concentrations across the study area, derived by subtracting the annual mean nitrate value for each year from the monthly nitrate concentrations. In most instances, nitrate anomalies exhibit consistency throughout the vertical profile, with this uniformity being more pronounced in the MED. Additionally, episodic discontinuities are often detected at depths around 100 m and approximately 500 m, corresponding to the mixing layer and certain pycnoclines. In the NEA, where seawater exchange is more dynamic, discontinuities in nitrate concentration are more prevalent.

This study identifies three significant temporal trends. Firstly, there is a discernible overall increase in nitrate concentrations, characterized by more frequent positive anomalies, particularly since 2021. This trend indicates an increasing fluctuation in nitrate levels and an escalation in eutrophication within the study area. Similar trends have been documented in studies employing BGC-Argo-based models (Fourrier et al., 2022) and those projecting nitrate dynamics based on RCP scenarios (Reale et al., 2022). Our findings extend these observations by providing results on a broader scale with more detailed quantification. Secondly, while vertical consistency of nitrate anomalies is stronger in the MED compared to the NEA, a decline in upper ocean nitrate concentrations is evident in the NEA. Ocean warming has hindered the upward transport of nutrient-rich cold water, modifying the vertical nitrate distribution in the NEA. Moreover, no consistent overall trend of anomalies is apparent in surface nitrate concentrations in the NEA, which may be driven by complex ocean-atmosphere interactions and anthropogenic



**Figure 11.** Interannual anomaly profiles of reconstructed nitrate concentrations in the MED (a) and NEA (b).

influences. Thirdly, the transition period of nitrate anomalies appears to be lengthening. The duration of individual positive or negative anomalies has extended from a few months at the beginning of the study period to several months or even over a year.

This lengthening may indicate irreversible shifts in the marine environment or a decline in the ocean's self-regulatory capacity.

Furthermore, interannual anomaly trends must be interpreted cautiously due to their dependence on SSEVs and specific weights of the model, and their reliability requires further research. For instance, the reconstructed results may overestimate



anomalies in the Bathypelagic Zone (>1000 m). At these depths, nitrate concentrations are relatively stable (Fig. 10) and are not effectively represented by SSEVs. Nonetheless, the model's estimates are inevitably influenced by the SSEV signal. The

accuracy of these anomalies is significantly influenced by both the generalization capacity of the model and the stability of the input features. Despite the model's proven reliability, particularly with the MEB performance most relevant to anomaly calculations consistently maintaining an excellent level of $\pm 0.04\ \mu mol \cdot kg^{-1}$ across multiple validations, small-scale findings still require further corroboration through targeted case studies. Overall, compared to the limitations posed by the discontinuous, discrete observations from BGC-Argo and the inadequacies of simulated nitrate in capturing fine-scale variability, the

reconstructed dataset offers an encouraging and comprehensive perspective for trend analysis.

### 3.6 Contribution of features to the model output

The current model is established based on all available features in the dataset, but given the redundancy among these features, the model may require fewer of the most efficient features. There are two benefits to entering all mentioned features. The ability of MLP to automatically extract features ensures that the model will be enhanced and minimally affected by feature redundancy.

Particularly, the utilization of large-scale simulated-nitrate data has significantly contributed to the model's capability to capture non-linear relationships among multiple features, thus enabling it to effectively monitor nitrate concentration across a wide range of SSEV scenarios. On the other hand, analyzing the importance of each feature based on the model with all the input features is crucial for further studies of nitrate estimation.

Given the high computational cost of SHAP (Chau et al., 2022) and the sufficient representativeness of a smaller sample

(Pauthenet et al., 2022), we randomly selected 50,000 samples to estimate feature contributions. Figure 12 shows the probability distribution and mean ASV for each feature, sorted by mean ASV. Given the different mechanisms by which features affect the ocean at various depths, the ocean is divided into two layers in the contribution discussion. Bounded by the 200 m depth, the upper layer is the Epipelagic Zone (EZ), and the deeper layer of 200-2000 m is the Mesopelagic Zone and part of the Bathypelagic Zone.

The input spatial features include depth, longitude, and latitude. Among them, depth is consistently identified as the strongest feature extracted by the model, which corresponds to the pattern of variations in the vertical distribution of nitrate. Particularly in the EZ, the contribution of depth features is extremely significant with $I_{\mathrm{Depth}}^{\mathrm{EZ}} = 2.41$ , surpassing that of other features and also much larger than the contribution of depth in the 200-2000 m depths. This is attributed to the fact that the increase in nitrate concentration with depth is most pronounced in the EZ. The nitrate concentration at 200 m depth may be several times

higher than that at the sea surface. Although such a trend is also observed in 200-2000 m depths, the magnitude of this change is relatively small. Hence, depth always remains the most crucial feature, especially in the EZ.

Furthermore, longitude is the second essential feature in the model. Nitrate concentration in the MED and NEA differs greatly, resulting in longitude more vital than latitude. The input temporal features include the cosine function Jday1 and the sine function Jday2 of the date. $I_{Jday1}$ is relatively larger, indicating that the cosine distribution of dates is more rewarding

for model prediction. In the 200-2000 m depths, the contribution proportion of spatiotemporal coordinates increase, while the contributions of other SSEV features decrease. On the one hand, nitrate concentration in the surface ocean is more susceptible



**Figure 12.** The probability distribution of the ASV, indicating the effect of each feature on the model output. The x-axis represents the ASV, scaled due to the large range, and the y-axis represents the input features, sorted by the total magnitude of the Ij. The filled area is the probability distribution of the ASV, and the labeled numbers are the mean ASV. The black vertical dashed lines represent the median and quartiles of the ASV.





to SSEVs. On the other hand, nitrate in the deep ocean exhibits low seasonal variability but stable regional characteristics, and its concentration is primarily related to its location. The estimation process in the deeper ocean mainly relies on spatiotemporal coordinates, supplemented by subtle adjustments in environmental variables.

The residual features comprise environmental parameters, which encompass diverse facets of climate, biology, and ocean dynamics. Among these parameters, SSH acquires the highest $I_j$. SSH reflects various dynamic effects of ocean circulation, mixing layers and eddies, which together influence the horizontal and vertical transport of nitrates (Ascani et al., 2013; Fripiat et al., 2021; Sarangi and Devi, 2017; Wang et al., 2021).

Another critical feature is SST, which previous research has indicated as the principal environmental factor for nitrate

retrieval. Upwelling and winter convective mixing constitute two crucial physical processes that drive the transportation of cold, nitrate-rich waters into the euphotic layer, thereby boosting SSN and simultaneously reducing SST (Kudela and Dugdale, 2000; Pan et al., 2018). Since SST and SSH provides information on vertical mixing, its contribution in the deep ocean remains significant compared to other SSEVs. Highly correlated features may dilute their individual contributions to the results, thus SST may have a more significant role than depicted in the figure. The variables PAR, ZEU and ZSD display a strong correlation

with SST and also exhibit elevated values of $I_j$ , indicating the dominant role of temperature-related features in the nitrate estimation process. In this case, ZEU and PAR are indicative of the optical environment as well as NLFH and CF, probably related to the oxidation of nitrogen by light inhibition and the activity of phytoplankton (Hutchins and Capone, 2022; Zakem et al., 2018). Ocean dynamics parameters (e.g., MLD and S10) also contribute to nitrate estimation by influencing nutrients in multiple ways (Tuerena et al., 2019; Liu et al., 2019).

As described above, the SHAP approach can explain the effect of each feature on the MLP output from both holistic and individual perspectives. This approach enables the comprehension of the role played by features in deep learning black box models. The ASV distribution of most features exhibits a long tail (Figure 12), suggesting that even features with low $I_j$ can have a significant impact on model estimation in extreme environments. Nevertheless, the SHAP contribution is solely based on mathematical models and data-driven interpretations, which may result in conclusions that contradict physical processes.

Although it has been experimentally validated that removing features with lower $I_j$ results in less decline in model accuracy, the evaluation of the features still requires caution. For instance, Chl has previously been utilized as another key variable to retrieve SSN (Goes et al., 1999; Pan et al., 2018; Sarangi and Devi, 2017), but the low contribution of Chl is not a fortuitous situation (Altieri et al., 2016) and its causes and mechanisms deserve further scrutiny and explanation.

## 4   Conclusions

This study developed a continual learning-based MLP model tailored to estimate the 3D ocean nitrate concentration. The model was cross-validated by independent in situ data profiles and demonstrated satisfactory performance, achieving an $R^2$ of 0.98, RMSE of 0.592 $\mu mol \cdot kg^{-1}$ and MAE of 0.398 $\mu mol \cdot kg^{-1}$. It also exhibited robust and reliable performance in both site-based cross-validation and independent cruise observations. Contrasting experiments show that the model's generalization is notably enhanced by employing continual learning from simulated-nitrate, especially in regions with limited data availability.



The estimation accuracy remains general stable across all dimensions, with the more significant error occurring within the vertical range of 60-100 m and in the sparse region of the observations.

The 3D spatiotemporal distribution of nitrate is analyzed based on the reconstruction results. The findings indicate a progressive increase in oligotrophy from the western to eastern regions of the study area. Nitrate concentration shows significant seasonal variability in the vertical dimension, driven by seawater exchange and biological processes. From an interannual
perspective, a discernible increase in nitrate concentrations was noted, especially since 2020. Besides, vertical consistency in interannual anomalies within the NEA was lacking, with discrepancies commonly observed around depths of 100 m and 500 m.

The contribution of each feature is calculated to gain insight into their influence on nitrate estimation. Results reveal that spatial coordinates such as depth, longitude and environmental variables represented by SSH and SST exert the most significant
influence. Meanwhile, certain features with low average contribution can still play vital roles in specific instances involving high anomalies.

From future perspectives, integrating remote sensing with deep learning for estimating oceanic 3D conditions holds significant research potential. Continual learning allows for the incorporation of numerical model knowledge to address the limitations of sparse in situ measurements and can be coupled with any deep learning model, making it a promising paradigm for recon-
structing ocean datasets. Given the challenges of continuously high-resolution ocean monitoring, this approach can serve as an alternative solution to bridge the observation gap. Leveraging remote sensing expands retrieved variables and adds vertical dimension insights, supporting further research into marine environment.

*Code availability.*

All code used in the current study is available from the corresponding author upon reasonable request.

*Data availability.*

The reconstructed 3D nitrate concentration dataset presented in the paper can be accessed via Zenodo at https://doi.org/10.5281/zenodo.14010813 (Yu et al., 2024). Here we provide nitrate concentration gridded product for the pan-European ocean at $0.25° \times 0.25°$ horizontal resolution on 63 vertical levels from 0–2000 m and at a monthly resolution from 2010 to 2023.

*Author contributions.*

XY: Methodology, Data Curation, Software, Visualization, Writing - original draft. HG: Conceptualization, Funding acquisition, Supervision. JZ: Conceptualization, Funding acquisition, Supervision, Writing - review & editing. YM: Methodology, Supervision. XW: Formal analysis, Validation, Writing - review & editing. GL: Data curation, Validation, Writing - review &

editing. MX: Formal analysis, Validation, Writing - review & editing. NX: Formal analysis, Validation, Writing - review & editing. AS: Formal analysis, Validation, Writing - review & editing. All authors have discussed the results and commented on
the manuscript.

*Competing interests.*

The contact author has declared that none of the authors has any competing interests

*Acknowledgements.* The authors thank the International Argo Program, European Space Agency, European Centre for Medium-Range Weather Forecasts (ECMWF), Copernicus Climate Change Service, and GLODAPv2 for data, which are freely accessible for public. The
BGC-Argo data were collected and made freely available by the International Argo Program and the national programs that contribute to it (https://argo.ucsd.edu, https://www.ocean-ops.org). The Argo Program is part of the Global Ocean Observing System. Biogeochemical data are from the European Copernicus Marine Environmental Monitoring Service (CMEMS, https://marine.copernicus.eu). The GlobColour data used in this study has been developed, validated, and distributed by ACRI-ST, France (http://globcolour.info). The ERA5 data are from the Copernicus Climate Change Service (https://cds.climate.copernicus.eu).

*Financial support.* The authors thank the anonymous reviewers whose comments and suggestions significantly improve this manuscript. This work was supported by the Strategic Priority Research Program of the Chinese Academy of Sciences-A (No. XDA19030402), Natural Science Foundation of China (No. 42071425), the "Taishan Scholar" Project of Shandong Province (No. TSXZ201712), and Discipline Cluster Research Project of Qingdao University "Deep mining and intelligent prediction of multimodal big data for marine ecological disasters" (No.XT2024101).



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
