# Peer review of "A continual learning-based multilayer perceptron for improved reconstruction of three-dimensional nitrate concentration"

_Earth System Science Data, 2024_

## Referee Comment (RC2)

The manuscript titled " A continual learning-based multilayer perceptron for improved reconstruction of three-dimensional nitrate concentration" presents a model to estimate the three-dimensional nitrate concentration in the Pan-European Ocean. Overall, this study is interesting, and the research and reconstruction data could contribute to a deeper understanding of regional ocean nutrient dynamics. The manuscript demonstrates good levels of research significance, innovation, and expression. However, before accept this work, there are some concerns need to be addressed.

Major Comments:

1. Sections 2.1: The study area is divided into two parts: MED and NEA. The observations in NEA are sparse. What is the rationale for selecting this region? How can the model's performance be ensured to remain stable in this area?

2. In Figure 6, the coupling of the continuous learning model improves overall performance. However, in terms of spatial distribution, errors increase at the locations of a few observation points. How should this phenomenon be understood, and is there potential for improvement? An additional discussion of the model's strengths and weaknesses is needed.

3. The simulated nitrate distribution pattern in Figure 6e appears somewhat unusual. How was the performance comparison of this widely used three-dimensional nitrate data product implemented here?

4. Regarding the interannual variation trend in Figure 11 and the abnormal increase after 2022, are there any studies or data that support similar conclusions? Caution is required when evaluating and interpreting these phenomena.

5. The feature importance calculated through SHAP includes both positive and negative values. The manuscript evaluates these values based on absolute values, which may overlook the sign of the contributions. Consider whether a deeper assessment of feature contributions using the positive and negative relationships of SHAP is possible.

Minor Comments:

1. The data processing steps and validation methods need more detailed explanation, such as interpolation for missing values and the division of the test set.

2. The manuscript's descriptions should be consistent, such as "Figure" or "Fig," and "pan-Europe" or "Pan-Europe."

3. The titles of Figures 6 and 7 are not sufficiently clear and would benefit from refinement and reorganization.

---

## Author Comment (AC1)

**The Response to Referee Comments**

Earth System Science Data,

Manuscript ID: essd-2024-508

**A continual learning-based multilayer perceptron for improved reconstruction of three-dimensional nitrate concentration**

Xiang Yu, Huadong Guo, Jiahua Zhang, Yi Ma, Xiaopeng Wang, Guangsheng Liu,Mingming Xing, Nuo Xu, and Ayalkibet M. Seka

Corresponding Author: Jiahua Zhang

zhangjh@radi.ac.cn

**Dear Editor and Referees,**

We appreciate your valuable feedback and insightful comments on our manuscript. Your suggestions have been immensely helpful in refining and improving our work, and they have provided significant guidance for our research. We have carefully addressed each comment and suggestion individually, providing detailed explanations and implementing necessary revisions accordingly.

For clarity, we have enclosed the reviewer comments in boxes with sequentially numbered, while our responses are presented in standard font. In our responses to referee comments, figure and table numbers are prefixed with 'R', whereas figures in the revised manuscript use standard numbering, and supplementary figures are marked with 'S'. We hope that our explanations and modifications sufficiently address your concerns and meet with approval.

We sincerely appreciate your time and effort in reviewing our manuscript and look forward to your feedback.

**The Referee #1's Comments**

**The Referee #1 General Comment**

The manuscript presents a multi-layered perceptron (MLP) model designed for continuous learning to enhance the three-dimensional reconstruction of nitrate concentrations in oceanic environments. This approach improves the model's generalization capabilities by incorporating the nonlinear relationships among various marine surface environmental variables and surface nitrate levels. The model has been successfully applied in the Mediterranean Sea and Northeast Atlantic. Additionally, the study provides a reconstructed nitrate dataset spanning from 2010 to 2023, which holds significant value for marine ecology and environmental research. Below are my comments and suggestions regarding the manuscript:

**Response:**

We sincerely appreciate your comprehensive evaluation of our manuscript. Your thorough review and insightful comments have been invaluable in enhancing the quality of our work. We have conducted extensive validations and discussions to carefully address your concerns and have made comprehensive revisions accordingly. We believe these modifications have strengthened the scientific rigor and clarity of our manuscript, and we hope they adequately resolve your concerns. We look forward to your feedback in the near future. Below, we provide detailed responses to each of your comments.

**Major Comments:**

**The Referee #1's Comment #1**

The methodology for cross-validation should be elaborated upon. The current description does not adequately clarify how cross-validation is conducted, nor does it explain the rationale for utilizing all Argo data in the validation process.

**Response:**

Thank you for your comments. We are pleased to have the opportunity to clarify the cross-validation methodology. In this study, a 5-fold cross-validation was employed on the BGC-Argo dataset to assess the model's performance. Specifically, the BGC-Argo dataset was evenly divided into five subsets, either based on a vertical observation cycle (as outlined in Section 3.1) or by individual site (as outlined in Section 3.2). During each training and validation iteration, one subset served as the test set, while the remaining four subsets were used for training. This process was repeated without overlap for five cycles, ensuring that each subset was used as the test set exactly once. This approach allows for the entire BGC-Argo dataset to be validated and statistical metrics to be evaluated. The primary advantage of this method is that it ensures each data point is independently used for both training and validation, thus minimizing the impact of data partitioning bias on performance evaluation. This methodology maximizes data utilization and provides a more robust assessment of the model's generalization capability. Additionally, we have enhanced the explanation in the revision (please also see Line 260-264).

**The Referee #1's Comment #2**

The rationale for the inclusion of numerous variables needs to be articulated. A robust artificial intelligence model should be grounded in physical principles; achieving favorable results without this foundation may hinder the accurate representation of dynamic processes. It is essential to conduct experiments to determine whether the exclusion of certain variables could enhance the simulation outcomes. Furthermore, the absence of consideration for influential factors such as precipitation and river runoff warrants explanation.

**Response:**

We are grateful for your invaluable comments, which have guided us to further examine the feature variables we employed. Determining the optimal combination of features for a deep learning model is indeed a challenging task, and we have made an effort to address this issue from the following perspectives.

Firstly, following with your comments, we systematically excluded variables and compared the cross-validation simulation results based on profiles, as shown in Table R1. Table R1 is sorted by the increase RMSE after the exclusion of each variable, with RMSE being the most crucial metric as it also serves as the loss function supervising the model's training. The RMSE increased to varying degrees after the exclusion of variables, leading to the inference that the initial feature combination is one of the most optimal. Depth, latitude, and longitude, the three spatial coordinates, remain the most significant variables, with the highest RMSE increase when excluded. In particular, depth is used not only as a feature, but more importantly to map depth profiles, so exclusions will show exaggerated variation. Figure R1 illustrates the performance of the test set after excluding several key variables.

The feature ranking in Table R1 closely aligns with that in Figure 13 of the manuscript, though there are some minor differences. Both discuss the importance of features, but the SHAP values in Figure 12 focus on the contributions of the features, while the RMSE changes in Table R1 emphasize the irreplaceability of features. For example, Z (Total terrain depth), which provides a unique perspective, has a small contribution but significantly impacts the model when excluded. Furthermore, when excluding features, we combined Jday1 (the cosine function of the Julian day) and Jday2 (the sine function of the Julian day), which have a high contribution but a minimal impact on model performance when excluded. This is because Jday is a heuristic feature that, while useful for providing temporal information, can be inferred through the periodic variation of other variables. Figure R3 shows the correlation heatmap between nitrate and all input variables. The current feature combination is sufficient and potentially redundant; some highly correlated features can substitute one another to some extent, which is why the RMSE increase after excluding high-contribution features like PAR is relatively small. However, the comparison experiments in Table R1 confirm that the model can accommodate these correlated features and enhance its performance.

As for factors like precipitation and river runoff, we have already considered total precipitation (TP) as a feature. However, river runoff is limited by the model's scale. Currently, our model uses environmental variables at the same spatial and temporal coordinates as the nitrate data, while the spatial location of river runoff differs from that of oceanic nitrate. While river runoff significantly affects oceanic nitrate, we aim to further investigate these features that are spatially distinct from nitrate in future studies. Meanwhile, coloured dissolved organic matter (CDOM) at 412 nm has been considered as a candidate variable for the presence of correlation in river runoff in previous studies [1], and the similar Coloured dissolved and detrital organic materials absorption coefficient (CDM) has been utilised in our features.

In summary, the inclusion of 22 variables in our model follows two guiding principles. First, the model aims to achieve optimal performance with this set of features, which, based on our current comparisons, represents

Table R1: (Table S1 in the Supplement) Model's 5-fold profile-based cross-validation performance after sequentially excluding variables, with variables sorted by the increase in RMSE.

| Excluded features | $R^2$ | RMSE | MBE | MAE | MedAE | Slope k | Intercept b | RMSE increase ratio |
|---|---|---|---|---|---|---|---|---|
| Original feature set | 0.98 | 0.592 | -0.04 | 0.398 | 0.265 | 0.967 | 0.222 | |
| Depth | 0.46 | 3.079 | 0.017 | 2.376 | 1.974 | 0.457 | 2.942 | 420.10% |
| Lat | 0.973 | 0.686 | 0.003 | 0.48 | 0.342 | 0.954 | 0.245 | 15.88% |
| Lon | 0.977 | 0.641 | 0.002 | 0.443 | 0.309 | 0.955 | 0.245 | 8.28% |
| SSH | 0.978 | 0.637 | -0.015 | 0.428 | 0.301 | 0.96 | 0.234 | 7.60% |
| ZSD | 0.977 | 0.632 | -0.016 | 0.442 | 0.314 | 0.958 | 0.246 | 6.76% |
| Z | 0.977 | 0.63 | -0.018 | 0.439 | 0.311 | 0.957 | 0.251 | 6.42% |
| ZEU | 0.978 | 0.629 | -0.013 | 0.438 | 0.309 | 0.957 | 0.248 | 6.25% |
| SST | 0.978 | 0.629 | -0.031 | 0.427 | 0.297 | 0.966 | 0.216 | 6.25% |
| CDM | 0.978 | 0.628 | -0.01 | 0.437 | 0.308 | 0.958 | 0.24 | 6.08% |
| ZHL | 0.978 | 0.627 | -0.007 | 0.438 | 0.31 | 0.957 | 0.241 | 5.91% |
| TP | 0.978 | 0.625 | -0.009 | 0.435 | 0.308 | 0.958 | 0.237 | 5.57% |
| V10 | 0.978 | 0.623 | -0.016 | 0.434 | 0.308 | 0.957 | 0.249 | 5.24% |
| S10 | 0.979 | 0.622 | -0.016 | 0.432 | 0.305 | 0.958 | 0.247 | 5.10% |
| U10 | 0.978 | 0.622 | -0.017 | 0.432 | 0.303 | 0.958 | 0.248 | 5.07% |
| CF | 0.978 | 0.621 | 0.002 | 0.431 | 0.304 | 0.96 | 0.218 | 4.90% |
| SP | 0.978 | 0.621 | -0.009 | 0.432 | 0.304 | 0.959 | 0.234 | 4.90% |
| Chl | 0.978 | 0.617 | -0.004 | 0.426 | 0.296 | 0.96 | 0.222 | 4.22% |
| MLD | 0.978 | 0.616 | -0.02 | 0.424 | 0.295 | 0.958 | 0.251 | 4.05% |
| NFLH | 0.979 | 0.61 | -0.023 | 0.419 | 0.289 | 0.961 | 0.237 | 3.04% |
| PAR | 0.979 | 0.609 | -0.024 | 0.419 | 0.29 | 0.961 | 0.239 | 2.87% |
| Jday | 0.98 | 0.596 | -0.024 | 0.407 | 0.28 | 0.966 | 0.212 | 0.68% |

one of the best feature combinations. Second, we strive to incorporate as many features and influencing factors previously used in marine nitrate estimation research as possible, ultimately discussing their contributions within the context of the study region and the employed model, such as variables like CDM [1], Chl [2], wind components (U10, V10, S10) [3], MLD [3], Jday [4], and so on. We have added Table R1 and Figure R1 as supplementary material and added discussion to the manuscript (please also see Line 543-553 in the revision), and Figure R2 has been added to the manuscript and discussed (please also see Line 513-521).

[Figure]

Figure R1: (Figure S3 in the Supplement) Scatter plot of the model's test set performance after excluding six typical features. The red line indicates the fitted trend of the data, while the black dashed line denotes the 1:1 parity line.

**The Referee #1's Comment #3**

The role of upper ocean phytoplankton should be critically evaluated, as they are significant sinks for nitrate. The analysis suggests that phytoplankton are not influential, which raises concerns regarding the validity of this finding. It is imperative to provide a detailed explanation of the model's capacity to capture the underlying physical and biological processes in the ocean. Is it possible that the inclusion of numerous other variables has overshadowed the impact of chlorophyll a (Chla)?

**Response:**

Thank you for your insightful comments. Phytoplankton is indeed a classic factor influencing nitrate and is a crucial parameter in its retrieval. Its contribution mechanism is of significant importance, and we have paid close attention to this issue, conducting in-depth research to support our conclusions in the following three aspects.

Firstly, the primary reason for the low contribution of Chl is the limitation of the spatial and temporal scope. Mechanistically, phytoplankton is stimulated by nitrate in the spring, consuming nitrate and increasing Chl, while in the autumn, the decay of phytoplankton leads to an increase in nitrate [2]. Although this relationship is strong, the effect is usually confined to areas where phytoplankton growth is vigorous (in

[Figure]

Figure R2: (Figure 14 in the Revision) Scatter plot of Chl contribution values across data samples. The x-axis represents SHAP contribution values, while the y-axis represents depth. The color of scatter points indicates the Chl feature values in each sample.

horizontal directions) and the upper layers of the ocean (in vertical directions), which constitute only a small portion of the ocean's three-dimensional field. Therefore, the influence of phytoplankton is diluted in ocean regions with low productivity or in the mid-to-lower ocean layers, where the contribution of phytoplankton is less significant. In this revision, we have performed more comprehensive sampling and calculations of SHAP contributions, providing a more stable evaluation of feature contributions. Figure R2 shows the distribution of Chl contribution among 200,000 samples, and Figure R9 (added as Figure 13 in the manuscript) presents the improved SHAP value distribution. Samples with low chlorophyll occupy the majority, with contributions clustered near zero, while high-Chl samples still provide significant contributions, increasing nitrate estimates by more than one unit (Figure R2). Compared to other variables, the frequency distribution of Chl contributions shows a more significant clustering around zero and a long tail for high values. Therefore, it can reasonably be inferred that Chl still plays an important role in nitrate estimation in the upper ocean layers when the feature value is high, but the low proportion of these data leads to a small average Chl contribution.

Secondly, as you mentioned, the inclusion of many other variables may somewhat obscure the impact of Chl. As shown in the correlation heatmap in Figure R3 and the contribution distribution in Figure R9, many variables highly correlated with Chl have high contributions, such as positively correlated CDM and negatively correlated ZHL and SST. Since the model's batch size for training is limited, with neuron weights adjusted by 256 data samples at each time, and considering the high proportion of low-Chl, low-contribution samples, the model may lean towards more generalized water color parameters.

Thirdly, similar conclusions regarding the low contribution of chlorophyll have been found in related studies. In another study on nitrate three-dimensional reconstruction, the contribution of Chl was also found to be low [5]. For the Indian Ocean, oceanic dynamics outweighed thermal factors, followed by biochemical factors (including Chl). In contrast, Chl typically shows a smaller contribution when using more parameters for three-dimensional estimation, as the spatial range is smaller, while it contributes more when fewer parameters are used for surface retrieval [1, 3, 2], as it provides features among strongly correlated surface biogeochemical parameters.

[Figure]

Figure R3: (Figure 12 in the Revision) Heatmap for the matrix of Pearson correlation coefficients between nitrate and input variables. The size of the cell represents the absolute value of the correlation coefficient

In conclusion, we believe that the relatively low average contribution of Chl is reasonable and well-founded. However, this is largely due to its low spatial coverage and does not contradict the mechanism that Chl is an important driver of nitrate. We have added Figure R2 to the revision (Figure 14) and comprehensively enhanced the discussion of Chl contribution (please also see Line 570-582 in the revision).

> **The Referee #1's Comment #4**
>
> A more comprehensive discussion of the model's limitations is necessary, particularly regarding the consistency and discrepancies between the model's results and existing literature, including its performance in data-sparse regions.

**Response:**

Thank you for your valuable comments. We will provide a more comprehensive discussion of the model.
A. discussion of the model's limitations
Our model has the following limitations: 1. Scale and Global Features: The model is based on water column profiles for estimation and reconstruction, rather than incorporating broader-scale global features. This may limit the model's potential, though this approach was driven by the sparse nature of BGC-Argo measurements and the costs associated with continual learning training. We aim to explore the application of global features in future research. 2. Continual Learning Strategy: The effectiveness of our continual learning strategy hinges on the relationship between the datasets, requiring a rich yet relatively accurate pretraining dataset

[Figure]

Figure R4: (Figure 6 in the Revision) Performance of the model in site-based cross-validation on the test set(a), spatial distribution (b), and model comparison (c-f). (c) shows the test performance without using continual learning, along with the spatial distribution of increased RMSE (d). (e) depicts the validation performance for simulated nitrate concentrations and the spatial distribution of increased RMSE compared to this model.

(CMEMS nitrate) and a precise but smaller final training dataset (BGC-Argo nitrate). Fortunately, the nitrate reconstruction task provides datasets that meet these needs and demonstrate the effectiveness of continual learning. 3. Training Balance in Well-Sampled Regions: After continual learning, the model may experience a slight decline in performance in the well-sampled BGC-Argo regions due to the retention of simulated nitrate knowledge. Although this adverse effect is minor and sparse (as shown in Figure R4), it may be addressed in future research to improve model performance. To the discussion of the characteristics of the model based on the experimental results in Chapter 3, we add a paragraph on the limitations of the model in the conclusion section (please also see Line 610-617).

B. the consistency and discrepancies between the model's results and existing literature

Our model achieves one of the highest levels of accuracy in current nitrate estimation studies and performs robustly in the multiple validations implemented. Given the differences in study regions and datasets, it is challenging to make a direct comparison between our model and existing literature. However, we have actually compared the model improvements through reproduction. The key innovation of our model lies in the use of continual learning, coupling numerical model nitrate knowledge to enhance generalization capability. The MLP model serves as a benchmark for this coupling and is widely used for 3D estimations of ocean parameters [6, 7]. Figure R4 (Figure 6 in the manuscript) shows the performance comparison of our model with continual learning. In the station validation, we achieved significant improvements: $R^2$ increased by 5.8% (from 0.928 to 0.877), and RMSE decreased by 23.3% (from 1.469 to 1.126), with regional RMSE differences plotted. This demonstrates the effectiveness of continual learning and highlights the differences compared to existing literature. Furthermore, continual learning, as a training strategy, has the potential to be integrated into more advanced models.

C. its performance in data-sparse regions

Regarding the model's performance in data-scarce regions, one of the reasons for choosing the Mediterranean and northeastern Atlantic (NEA) as our study area was to verify whether effective global modeling could be achieved in the context of data imbalance and environmental differences. NEA accounts for about 14% of the dataset, but it is located in a vast and dynamic marine environment, making nitrate estimation more challenging. In the cross-validation implemented in Sections 3.1 and 3.2, NEA showed reasonably robust estimation performance, though slightly weaker than MED.

Currently, temporally and spatially continuous nitrate datasets available for comparison are scarce, with the CMEMS nitrate dataset being one of the most accessible. Its spatial distribution is illustrated in Figure R5. Our reconstructed results exhibit strong spatial consistency with CMEMS nitrate data, including in regions not covered by BGC-Argo, such as the coastal areas of the UK and Norway. Furthermore, our approach is primarily designed to validate model performance in data-sparse regions using an independent nitrate measurement dataset, GLODAPv2, which focuses on the NEA, where BGC-Argo coverage is particularly weak. As demonstrated in the validation presented in Section 3.3, the model maintains high estimation accuracy for samples from this dataset and outperforms CMEMS nitrate data in comparative assessments. Overall, the model's performance in data-scarce regions is reliable and surpasses the performance of current numerical models and deep learning methods. We have added Figure R5 to the Supplement (Figure S1) and discussed extensively the performance of the model in data-sparse regions in Sections 3.2-3.4.

We hope these clarifications address your concerns, and we have enriched the manuscript discussion based on the above points.

**Minor Comments:**

> **The Referee #1's Comment #5**
>
> Line 5: Provide the full name for "CMEMS".

**Response:**

Thank you for your careful review. The full name of CMEMS is the Copernicus Marine Environment Monitoring Service, and we have updated the manuscript accordingly (please also see Line 5).

[Figure]

Figure R5: (Figure S1 in the Supplement) Spatial distribution of CMEMS simulated nitrate field, with columns representing four seasons and rows representing five depth slices.

**The Referee #1's Comment #6**

Line 36: Provide the full name for "SSN"

**Response:**

Thank you for your comment. We have provided the full name for "SSN" in Line 36. The acronym "SSN" stands for sea surface nitrate, and we have updated the manuscript accordingly (please also see Line 36).

**The Referee #1's Comment #7**

Line 45: Several studies, for example, Liu et al. (2022, http://doi.org/10.3390/rs14195021) have considered other factors besides SST.

**Response:**

Thank you for your thorough review and insightful comments. We have incorporated the references and made a careful comparison. Besides SST, Chl-a, and spatio-temporal coordinates, Liu et al. [8] used two features, the diffuse attenuation coefficient at 490 nm (Kd490) and sea surface salinity (SSS), but these two features were limited in our study.

For Kd490, the observing method leads to a significant amount of missing values in high latitude regions during winter. Figure R6 visualizes the Kd490 data product for December 2023, derived from the GlobColour platform (`https://hermes.acri.fr/`), showing that data are missing for areas north of about 47°N. Therefore, using Kd490 as a feature within our study region (30°N-65°N) would limit the continuous estimation of nitrate.

[Figure]

Figure R6: Visualization of the Kd490 data product for December 2023.

[Figure]

(a) Original feature combination in the manuscript
(b) Feature combination with SSS

Figure R7: Comparison of test set performance in profile-based cross-validation with the inclusion of SSS feature.

For SSS, this data product carries substantial uncertainty, which could adversely affect the model's feature set. SSS data is typically derived from biogeochemical models, numerical models, or interpolation methods, rather than direct remote sensing observations. We tested the CMEMS salinity product (`https://doi.org/10.48670/moi-00021`). Figure R7 presents a performance comparison between the feature set used in the manuscript and the inclusion of SSS, with the results indicating a slight decline in performance upon adding SSS. Additionally, since both the pre-trained nitrate and salinity data come from similar CMEMS numerical models and are highly correlated, the uncertainty within the data may interfere with the model's ability to learn generalized knowledge [6]. Therefore, while SSS is a highly promising feature, its current uncertainty may have detrimental effects, limiting its application in our model.

> **The Referee #1's Comment #8**
>
> Line 81-82: Need more explanation.

**Response:**

Thank you for your detailed comments and we are pleased to provide more explanation on this issue.The original text of lines 81-82 is: "*However, this study relied on simulated data for supervised training instead of actual observational data, which may limit the model's applicability in real ocean environments.*"

Yang et al.[5] used two deep learning models to reconstruct three-dimensional nitrate in the Indian Ocean, providing invaluable studies with great results and insightful findings. However, these models require capturing vast spatial features and a substantial amount of supervision, which makes it challenging for in-situ measurements like BGC-Argo data to meet these needs. Their nitrate training data relied on the CMEMS nitrate product(`https://data.marine.copernicus.eu/product/GLOBAL_MULTIYEAR_BGC_001_029`), which is the same pre-trained nitrate data used in our study. This product still differs considerably from actual ocean conditions, especially in its inability to capture mesoscale variations effectively, as highlighted in both

Figure 10 of the literature [5] and Figure R4e (also Figure 6e in our manuscript). Therefore, we summarized the current state and limitations of research in lines 81-82 to acknowledge this gap.
* * *
**The Referee #1's Comment #9**

Line 94: "Pan-European" or "pan-European"? should be identical.

**Response:**

Thank you for your meticulous review and for helping us refine the manuscript. We have standardized the term "pan-European" throughout the manuscript as per your suggestion.
* * *
**The Referee #1's Comment #10**

Line 95: Please check the definition of "shelf sea". Apparently, the study area is beyond the shelf sea.

**Response:**

Thank you for your thoughtful comments and for highlighting the importance of using precise terminology when describing the study area.

You are absolutely correct that the study area extends beyond the shelf-sea regions. Initially, we selected this region for its rich BGC-Argo data and its proximity to land compared to other data-dense areas, which provided a solid foundation for our work. Then, as the research progressed, we were able to expand the study area to include more open ocean regions, leading to more comprehensive results.

We recognize that the current description of the study area may not fully reflect its diverse scope, so we have revised the manuscript accordingly. The updated description now clearly states that the study area encompasses key shelf-sea regions along the European coast but also extends to the open ocean beyond the continental shelf. We have toned down the emphasis on shelf-sea regions while retaining the two key reasons for choosing this area: the availability of BGC-Argo data and the need to focus on nitrate dynamics in marine regions closer to land. We have revised this part of the description for greater clarity (Lines 95-98).

We appreciate your constructive feedback and believe these revisions make the manuscript clearer. If you have any further suggestions, please let us know.
* * *
**The Referee #1's Comment #11**

Section 2.2: There is no introduction to the CMEMS simulated nitrate data, which should be introduced detailedly.

**Response:**

We appreciate your careful review that guided us in refining the manuscript. We have added Section 2.2.2 Simulated nitrate data for the presentation of simulated nitrate data from CMEMS (please also see Line 115-129)

**The Referee #1's Comment #12**

Figure 3: It looks familiar to me, and although the author may have redrawn it from a certain document, the original source should be noted.

**Response:**

Thank you for your comments and we appreciate your attention to detail. Figure 3 is a reference to the EWC literature for reflecting the algorithmic ideas of EWC [17]. The figure was redrawn to incorporate the nitrate dataset relationships, and we have now added citations in both the title note and the main text.

**The Referee #1's Comment #13**

Line 274: ". the" should be ". The".

**Response:**

Thank you for pointing that out. We have corrected the capitalization in Line 274 (now Line 295 in the revision), changing ". the" to ". The" to ensure proper sentence structure and consistency throughout the manuscript.

**The Referee #1's Comment #14**

"Figure" or "Fig. " should be identical throughout the text.

**Response:**

We greatly appreciate your attention to the details of our manuscript, which has helped improve many aspects of the work. We did notice that the ESSD journal has specific submission guidelines regarding figure numbering (`https://www.earth-system-science-data.net/submission.html#figurestables`):

*The abbreviation "Fig." should be used when it appears in running text and should be followed by a number unless it comes at the beginning of a sentence, e.g.: "The results are depicted in Fig. 5. Figure 9 reveals that...".*

We have thoroughly reviewed all figure numbering throughout the manuscript and will double-check this issue again to ensure compliance. We hope this helps address your concerns.

**The Referee #1's Comment #15**

Line 317-318: More introductions need to explain why and how.

**Response:**

Thank you for your review and comments. We would like to provide a more detailed explanation of this issue.

The original text of lines 317-318 is: "*In this section, a more rigorous cross-validation is implemented, which divides the dataset based on the BGC-Argo device, as shown in Fig. 6.*"

In 3D reconstruction, the most crucial factor is the model's robust generalization ability. This is because there is always limited temporal and spatial coverage by measurement values, and maintaining stable estimates in areas not covered by measurements is key to obtaining a reliable 3D nitrate dataset. Therefore, the stringency of cross-validation, in terms of validating the model's generalization capacity, depends on the degree of difference between the training and testing datasets. In Section 2.3.4, we describe the approach of dividing the dataset based on BGC-Argo profiles (implemented in Section 3.1) and stations (implemented in Section 3.2). A profile represents a single measurement cycle of BGC-Argo, with validation carried out in Section 3.1. A station, on the other hand, represents a BGC-Argo device, and there are significant temporal and spatial differences in nitrate measurements across different stations. This division is evidently more varied than that based on profiles, which is why the description in lines 317-318 of the original text appears in Section 3.2. Under this more stringent validation approach, which challenges the model's generalization ability, the improvements introduced by our model become more pronounced. We have revised this section to more clearly describe the validation approach and differences (please also see Line 338-342 in the revision).
* * *
Thank you for your thoughtful review and valuable comments. We have carefully considered your suggestions and made revisions to improve the clarity and accuracy of the manuscript. Your thorough review has greatly contributed to enhancing the quality of the manuscript, and we hope these revisions meet your expectations. If you have any further suggestions, please let us know.

**The Referee #2's Comments**

**The Referee #2's General Comment**

The manuscript titled " A continual learning-based multilayer perceptron for improved reconstruction of three-dimensional nitrate concentration" presents a model to estimate the three-dimensional nitrate concentration in the Pan-European Ocean. Overall, this study is interesting, and the research and reconstruction data could contribute to a deeper understanding of regional ocean nutrient dynamics. The manuscript demonstrates good levels of research significance, innovation, and expression. However, before accept this work, there are some concerns need to be addressed.

**Response:**

We sincerely appreciate your comprehensive evaluation of our manuscript. Your detailed review and insightful comments have been instrumental in improving the quality of our work. We believe that these revisions have enhanced the scientific rigor and clarity of our manuscript, and we hope they adequately address your concerns and gain your approval. Below, we provide detailed responses to each of your comments.

**Major Comments:**

**The Referee #2's Comment #1**

Sections 2.1: The study area is divided into two parts: MED and NEA. The observations in NEA are sparse. What is the rationale for selecting this region? How can the model's performance be ensured to remain stable in this area?

**Response:**

Thank you for your insightful comments. We greatly appreciate the opportunity to further clarify our motivations in detail.

The NEA region accounts for only about 14% of BGC-Argo measurements and presents a more complex oceanic environment in open waters, making nitrate estimation particularly challenging. This challenge is exacerbated by the significant differences in data distribution between the Northeast Atlantic (NEA) and the Mediterranean Sea (MED) measurement labels during cross-validation. However, NEA is a key region of ocean productivity with high scientific significance. One of our primary motivations is to develop a consistent model capable of reliably estimating three-dimensional nitrate concentrations across diverse oceanic conditions. Therefore, we included NEA in our study, treating it as a hypothetical data-sparse region to test the generalization capability of our model. To ensure robust validation, we conducted region-specific performance evaluations (Figures 4–7 in the manuscript) and achieved promising results. We have described the reasons for selecting the study area more clearly in our revision (please also see Line 99-102).

To maintain stable performance in data-sparse regions such as NEA, we developed a continual learning strategy by leveraging relationships within existing datasets to enhance the model's generalization capability. Specifically, although BGC-Argo measurements are sparse in NEA, ocean surface features and CMEMS nitrate data remain globally continuous and available. By pretraining on CMEMS nitrate distributions and refining the model through a second training phase with BGC-Argo data, we effectively improved its generalization ability. The detailed algorithmic approach can be found in Section 2.3.3. We hope this explanation

addresses your concerns.

**The Referee #2's Comment #2**

In Figure 6, the coupling of the continuous learning model improves overall performance. However, in terms of spatial distribution, errors increase at the locations of a few observation points. How should this phenomenon be understood, and is there potential for improvement? An additional discussion of the model's strengths and weaknesses is needed.

**Response:**

Thank you for your meticulous review and insightful observations. In the results of Figure 6, a slight increase in error is observed at certain horizontal locations in a few test sets (though the magnitude of increase is minor, as the negative blue portion of the color scale has been magnified). Beyond the natural fluctuations in model performance across different validations, this phenomenon is primarily attributed to the balance of knowledge between the CMEMS nitrate and BGC-Argo nitrate datasets.

Figure 3 in the manuscript illustrates this from an algorithmic perspective — continual learning enables the model to maintain distributional knowledge from both nitrate datasets simultaneously. However, in regions where BGC-Argo sampling is sufficiently dense, closely spaced BGC-Argo stations may be allocated to both the training and test sets. In such cases, the observed measurements are already sufficient to support highly accurate model estimates, while distribution and uncertainty in numerical models (as shown in Figure 6e of the manuscript) may introduce some interference. Nevertheless, fully sampled BGC-Argo regions account for only a small proportion of the dataset, while in most test sets, error reductions are predominantly observed, demonstrating significant improvements in regions previously not covered by BGC-Argo.

This phenomenon presents an opportunity for further improvement. The parameter $\lambda$ in Equation (11) is designed to balance the knowledge contribution of CMEMS and BGC-Argo nitrate datasets. By lowering $\lambda$, the model can place greater emphasis on BGC-Argo nitrate knowledge, leading to more accurate estimates in well-sampled regions. Currently, $\lambda$ is set to $10^7$, based on achieving the optimal overall performance across test sets. In future research, we plan to introduce a dynamic $\lambda$ setting or an additional module to adapt varying sampling densities across regions and mitigate this limitation.

In summary, the strengths and limitations of the model are intuitively demonstrated in Figure 3 of the manuscript. The primary advantage is that the model, through continual learning, integrates CMEMS nitrate knowledge, significantly enhancing computational performance and generalization in data-sparse regions, leading to an overall substantial reduction in estimation errors. The limitation lies in the potential uncertainty introduced by continual learning in fully sampled regions, but this presents an avenue for improvement in future studies. In the revision, we discuss the strengths of the model in the analysis of the results of multiple experiment results and provide additional discussion of the limitations of the model (please also see Line 610-617 in the revision).

**The Referee #2's Comment #3**

The simulated nitrate distribution pattern in Figure 6e appears somewhat unusual. How was the performance comparison of this widely used three-dimensional nitrate data product implemented here?

**Response:**

Thank you for your detailed comments. Figure 6e in the manuscript compares the simulated nitrate concentrations from CMEMS with in-situ nitrate measurements from BGC-Argo to assess the accuracy of the former. Specifically, CMEMS provides a gridded dataset of nitrate concentrations with a spatial resolution of 0.25° in longitude and latitude, 75 vertical depth levels (54 within 0–2000 m), and a temporal resolution of monthly averages. In contrast, BGC-Argo measurements do not adhere to a standardized resolution but instead record the precise longitude, latitude, depth, and timestamp of each observation. To approximate the simulated values, CMEMS grid data were interpolated along four dimensions (longitude, latitude, depth, and time) to match the spatiotemporal coordinates of BGC-Argo observations, and the comparison results are presented in Figure 6e.

Regarding the distribution pattern observed in Figure 6e, particularly the stepwise clustering of scatter points, we provided an alternative comparison in Figure 8e of the manuscript, which presents a seasonal-depth profile. One key limitation of CMEMS nitrate data is its lack of variability in capturing localized vertical dynamics in the mid-ocean layers. As a result, multiple depth levels within a certain range are assigned similar concentration values. This phenomenon manifests in Figure 6e as the formation of horizontal stepwise clusters, where segments exhibiting gradients in nitrate concentration along the x-axis correspond to near-identical values along the y-axis. The revisions were made in the manuscript, please also see Line 343-347.

> ### The Referee #2's Comment #4
>
> Regarding the interannual variation trend in Figure 11 and the abnormal increase after 2022, are there any studies or data that support similar conclusions? Caution is required when evaluating and interpreting these phenomena.

**Response:**

We sincerely appreciate your valuable comments. Investigating global three-dimensional nitrate trends from a quantitative perspective remains a significant challenge, and addressing this issue is a key contribution of our study. Currently, there is no clear and unified conclusion regarding oceanic three-dimensional nitrate trends, but our analysis has received support from both data and literature perspectives, as outlined below.

From the data perspective, the most comprehensive and promising datasets for describing three-dimensional nitrate distributions are the BGC-Argo and CMEMS nitrate datasets, which we employed in our study, representing observational and modeled approaches, respectively. Figure R8 illustrates the interannual trends of these datasets in the MED and NEA regions. However, both datasets have significant limitations when depicting interannual trends. Due to the varying geographical locations of BGC-Argo observations over time, regional differences in nutrient levels introduce considerable interference and fluctuations in interannual trend calculations. As a result, BGC-Argo trends tend to be more pronounced, and in some cases, shifting sampling locations across different nutrient regimes may even lead to trend reversals. Conversely, CMEMS nitrate data exhibit a delayed response in capturing mesoscale nitrate variability, as noted in your third comment. Therefore, the overly uniform trend patterns observed in Figure R8bd are likely more conservative than actual conditions, yet they still provide a useful reference for decadal-scale trends.

From the literature perspective, our analysis aligns with certain previous studies. While some theories suggest that enhanced ocean stratification due to climate warming weakens upper-layer nutrient availability, recent research indicates that this effect is more pronounced for phosphate, whereas nitrate continues to exhibit frequent regional variability [18]. Our reconstruction results reveal pronounced interannual variability in

(a) Argo for MED

(b) Argo for NEA

(c) CMEMS for MED

(d) CMEMS for NEA

Figure R8: (Figure S2 in the Supplement) Interannual trends in nitrate from multi-source datasets

the NEA, with the 2010–2014 upper-ocean positive anomalies [19] and the overall increasing trend in the Iberian Upwelling System [20] being supported by previous studies. In contrast, interannual anomalies in the MED appear more regular and are supported by a greater number of case studies. In the Sicily Channel, nitrates showed a slightly negative trend from 2011 to 2016, which then turned positive until 2020 [21]. In the northwestern MED, nutrient concentrations exhibited an overall increasing trend from 2013 to 2020, with deep-ocean nitrate accumulation being more pronounced than in the upper layers [22]. The general trends observed in the MED are consistent with ocean environmental projections simulated using physical-biogeochemical models under Representative Concentration Pathways (RCPs) 4.5 and 8.5 [23]. Notably, the anomalous increase observed after 2022 may be linked to the intense winter storm Carmel in 2021. A four-year time-series study at a Levantine Basin site indicates that the declining oceanic nitrate levels since 2018 were significantly replenished during the 2021 winter storm [24].

In summary, given the ongoing uncertainties surrounding 3D ocean nitrate trends, our analysis is supported by both data and literature. Moreover, our study provides one of the most detailed quantitative analyses available, offering a unique perspective and deeper understanding of this issue. We have carefully evaluated this critical scientific question and remain cautiously optimistic about our findings, aiming to contribute a distinct viewpoint and help bridge existing observational gaps. We have incorporated Figure R8 into the Supplementary Material (Figure S2) and have enhanced the discussion of reference citations and trends across the board in our revisions (please also see Line 462-488). Thank you for your thorough review, and we hope these clarifications address your concerns.

**The Reviewer #2's Comment #5**

The feature importance calculated through SHAP includes both positive and negative values. The manuscript evaluates these values based on absolute values, which may overlook the sign of the contributions. Consider whether a deeper assessment of feature contributions using the positive and negative relationships of SHAP is possible.

**Response:**

We sincerely appreciate your invaluable insights, which have inspired and guided us in refining our work. As you suggested, the positive and negative relationships in SHAP values can be leveraged for a more in-depth assessment of feature contributions. The sign of the SHAP contribution value indicates the direction of its impact on the estimation results, while the absolute value represents the magnitude of this impact. Considering the initial SHAP values allows for a more comprehensive evaluation of feature contributions. However, since the presence of both positive and negative initial SHAP values can interfere with the ranking of average feature contributions, we have adopted a combined approach that simultaneously considers both the initial SHAP values and their absolute values.

To address this, we increased the sampling ratio of SHAP contributions during the manuscript revision process, enabling a more thorough computation. We have updated the SHAP contribution distribution diagram to Figure R9 (please also see Figure 13 in the revision) and revised the discussion in Section 3.6 accordingly, please see manuscript for details (Line 513-583). Compared to our original approach, your suggestion allows for a richer perspective in analyzing feature contributions from multiple angles, particularly for variables like depth, where distinct patterns emerge between the upper and deeper ocean layers.

**Minor Comments:**

**The Reviewer #2's Comment #6**

The data processing steps and validation methods need more detailed explanation, such as interpolation for missing values and the division of the test set.

**Response:**

We sincerely appreciate your valuable suggestions. We recognize the need for clearer articulation of data processing and validation methods and have provided more detailed clarifications and revisions accordingly.

1. Interpolation for Missing Values: Due to the high latitude of the study region, the observation of certain remote sensing parameters is limited. We addressed this issue by selecting well-available feature parameters, avoiding those with excessive missing values, such as Kd490 in Figure R6. The purpose of data preprocessing was to standardize the feature dataset and resample it to the specific spatiotemporal coordinates of BGC-Argo. For the minor missing values, we first applied linear interpolation to the gridded feature dataset in a sequential manner—longitude, latitude, and time. This approach helped fill in localized gaps while excluding large-scale missing data, thereby striking a balance between preserving useful information and reducing potential bias. Afterward, the interpolated gridded feature set was linearly interpolated and resampled to match the spatiotemporal coordinates of BGC-Argo and CMEMS nitrate data. During this process, we excluded samples where BGC-Argo coordinates were near grid features with missing values to prevent low-quality data from negatively impacting model training. We have provided a detailed explanation in the manuscript, please also

[Figure]

Figure R9: (Figure 13 in the Revision) Probability distribution of SHAP values representing the impact of each feature on the model output. The y-axis shows the input features, sorted by the total magnitude of $I_j$, while shaded area in the x-axis direction represents the distribution of SHAP values, scaled due to the large range. The numbers labeled on the left show the mean of the raw SHAP values, while those on the right show the mean of the ASV. The black vertical dashed lines represent the median and quartiles of the SHAP values.

2. Division of the Test Set. In this study, we implemented a five-fold cross-validation approach to assess the performance of our model on BGC-Argo data. Specifically, the BGC-Argo dataset was divided based on vertical observation cycle profiles (as described in Section 3.1) or individual sites (as described in Section 3.2). These units were then evenly split into five subsets. During each training and validation iteration,

one subset was designated as the test set, while the remaining four subsets were used for training. This process was repeated five times, ensuring that each subset served as the test set once. By doing so, the entire BGC-Argo dataset was systematically validated, and statistical metrics were evaluated comprehensively. The key advantage of this approach is that it guarantees each data point is independently utilized for both training and validation, thereby minimizing the impact of data partition bias on performance evaluation. This maximizes data utilization and enables a more robust assessment of the model's generalization capability. We have provided a detailed explanation in the manuscript, please also see Line 261-265 in the revision.

> **The Referee #2's Comment #7**
>
> The manuscript's descriptions should be consistent, such as "Figure" or "Fig," and "pan-Europe" or "Pan-Europe."

**Response:**

Thank you for your thorough review, which has been invaluable in enhancing the quality of our manuscript. Based on your comments, we tandardized the region to "pan-European", checked and revised the figure titles according to the ESSD submission guidelines (`https://www.earth-system-science-data.net/Submission.html#figurestables`), and standardized the figure titles at the beginning of a sentence to "Figure" and in the middle of a sentence to "Fig.". Furthermore, we have conducted a comprehensive consistency check throughout the manuscript. We greatly appreciate your help.

> **The Referee #2's Comment #8**
>
> The titles of Figures 6 and 7 are not sufficiently clear and would benefit from refinement and reorganization.

**Response:**

Thank you for your helpful suggestion regarding the figure titles. We have improved them to make the descriptions more accurate and easier to understand, please also see Line 387, 419 in the revision. The revised titles now better reflect the content and focus of each chart.
* * *
We once again thank you for your thoughtful review and valuable comments, which have significantly helped us improve the manuscript. We have carefully followed your suggestions and strengthened our discussion to enhance the manuscript's accuracy and scientific rigor. We hope these responses and revisions meet your expectations and address your concerns. If you have any further suggestions, please feel free to let us know.

**References**

[1] Xiaoju Pan, George TF Wong, Tung-Yuan Ho, Jen-Hua Tai, Hongbin Liu, Juanjuan Liu, and Fuh-Kuo Shiah. Remote sensing of surface [nitrite+ nitrate] in river-influenced shelf-seas: The northern South China Sea Shelf-sea. *Remote Sens. Environ.*, 210:1–11, 2018. `doi:10.1016/j.rse.2018.03.012`.

[2] Joaquim I. Goes, Toshiro Saino, Hiromi Oaku, and Ding Long Jiang. A method for estimating sea surface nitrate concentrations from remotely sensed SST and chlorophyll aa case study for the north Pacific Ocean using OCTS/ADEOS data. *IEEE Transactions on Geoscience and Remote Sensing*, 37(3):1633–1644, 1999. `doi:10.1109/36.763279`.

[3] Xiaolei Yu, Shuangling Chen, and Fei Chai. Remote Estimation of Sea Surface Nitrate in the California Current System From Satellite Ocean Color Measurements. *IEEE Transactions on Geoscience and Remote Sensing*, 60(99):1–17, 2022. `doi:10.1109/TGRS.2021.3095099`.

[4] M. Sammartino, BB Nardelli, S. Marullo, and R. Santoleri. An Artificial Neural Network to Infer the Mediterranean 3D Chlorophyll-a and Temperature Fields from Remote Sensing Observations. *Remote Sensing*, 12:4123, 2020. `doi:10.3390/rs12244123`.

[5] Guangyu Gary Yang, Qishuo Wang, Jiacheng Feng, Lechi He, Rongzu Li, Wenfang Lu, Enhui Liao, and Zhigang Lai. Can three-dimensional nitrate structure be reconstructed from surface information with artificial intelligence? — a proof-of-concept study. *Science of the Total Environment*, 924:171365, May 2024. `doi:10.1016/j.scitotenv.2024.171365`.

[6] Tian Tian, Lijing Cheng, Gongjie Wang, John Abraham, Wangxu Wei, Shihe Ren, Jiang Zhu, Junqiang Song, and Hongze Leng. Reconstructing ocean subsurface salinity at high resolution using a machine learning approach. *Earth Syst. Sci. Data*, 14(11):5037–5060, November 2022. `doi:10.5194/essd-14-5037-2022`.

[7] Lixin Wang, Zhenhua Xu, Xiang Gong, Peiwen Zhang, Zhanjiu Hao, Jia You, Xianzhi Zhao, and Xinyu Guo. Estimation of nitrate concentration and its distribution in the northwestern Pacific Ocean by a deep neural network model. *Deep Sea Research Part I: Oceanographic Research Papers*, 195:104005, May 2023. `doi:10.1016/j.dsr.2023.104005`.

[8] Hao Liu, Lei Lin, Yujue Wang, Libin Du, Shengli Wang, Peng Zhou, Yang Yu, Xiang Gong, and Xiushan Lu. Reconstruction of Monthly Surface Nutrient Concentrations in the Yellow and Bohai Seas from 2003–2019 Using Machine Learning. *Remote Sensing*, 14(19):5021, 2022. `doi:10.3390/rs14195021`.

[9] Alan Longhurst, Shubha Sathyendranath, Trevor Platt, and Carla Caverhill. An estimate of global primary production in the ocean from satellite radiometer data. *J. Plankton Res.*, 17(6):1245–1271, 1995. `doi:10.1093/plankt/17.6.1245`.

[10] Jason Holt, James Harle, Roger Proctor, Sylvain Michel, Mike Ashworth, Crispian Batstone, Icarus Allen, Robert Holmes, Tim Smyth, and Keith Haines. Modelling the global coastal ocean. *Philos. T. Roy. Soc. A.*, 367(1890):939–951, 2009. `doi:10.1098/rsta.2008.0210`.

[11] S. V. Smith and J. T. Hollibaugh. Coastal metabolism and the oceanic organic carbon balance. *Rev. Geophys.*, 31(1):75–89, 1993. `doi:10.1029/92rg02584`.

[12] J.-P. Gattuso, Michel Frankignoulle, Roland Wollast, and Systematics. Carbon and carbonate metabolism in coastal aquatic ecosystems. *Annu. Rev. Ecol.*, 29(1):405–434, 1998. `doi:10.1146/annurev.ecolsys.29.1.405`.

[13] Sara Berglund, Kristofer Döös, Sjoerd Groeskamp, and Trevor McDougall. North Atlantic Ocean Circulation and Related Exchange of Heat and Salt Between Water Masses. *Geophys. Res. Lett.*, 50(13):e2022GL100989, July 2023. `doi:10.1029/2022GL100989`.

[14] C. M. Moore, M. M. Mills, K. R. Arrigo, I. Berman-Frank, L. Bopp, P. W. Boyd, E. D. Galbraith, R. J. Geider, C. Guieu, and S. L. Jaccard. Processes and patterns of oceanic nutrient limitation. *Nature geoscience*, 6(9):701–710, 2013. `doi:10.1038/ngeo1765`.

[15] Olivier Aumont, Christian Éthé, Alessandro Tagliabue, Laurent Bopp, and Marion Gehlen. PISCES-v2: An ocean biogeochemical model for carbon and ecosystem studies. *Geosci. Model Dev. Discuss.*, 8(2):1375–1509, 2015.

[16] G. Madec. NEMO reference manual 3_6_STABLE: NEMO ocean engine. *Note du Pôle modél. Inst. Pierre-Simon Laplace (IPSL) Fr.*, 2016.

[17] James Kirkpatrick, Razvan Pascanu, Neil Rabinowitz, Joel Veness, Guillaume Desjardins, Andrei A. Rusu, Kieran Milan, John Quan, Tiago Ramalho, Agnieszka Grabska-Barwinska, Demis Hassabis, Claudia Clopath, Dharshan Kumaran, and Raia Hadsell. Overcoming catastrophic forgetting in neural networks. *Proceedings of the National Academy of Sciences of the United States of America*, 114(13):3521–3526, March 2017. `doi:10.1073/pnas.1611835114`.

[18] Skylar D. Gerace, Jun Yu, J. Keith Moore, and Adam C. Martiny. Observed declines in upper ocean phosphate-to-nitrate availability. *Proceedings of the National Academy of Sciences*, 122(6):e2411835122, February 2025. URL: `https://pnas.org/doi/10.1073/pnas.2411835122`, `doi:10.1073/pnas.2411835122`.

[19] V. A. Macovei, S. Torres-Valdés, S. E. Hartman, U. Schuster, C. M. Moore, P. J. Brown, D. J. Hydes, and R. J. Sanders. Temporal variability in the nutrient biogeochemistry of the surface north atlantic: 15 years of ship of opportunity data. *Global Biogeochemical Cycles*, 33(12):1674–1692, 2019. `doi:10.1029/2018GB006132`.

[20] Xosé Antonio Padin, Antón Velo, and Fiz F. Pérez. ARIOS: A database for ocean acidification assessment in the iberian upwelling system (1976–2018). *Earth System Science Data*, 12(4):2647–2663, November 2020. `doi:10.5194/essd-12-2647-2020`.

[21] Francesco Placenti, Marco Torri, Federica Pessini, Bernardo Patti, Vincenzo Tancredi, Angela Cuttitta, Luigi Giaramita, Giorgio Tranchida, and Roberto Sorgente. Hydrological and biogeochemical patterns in the sicily channel: New insights from the last decade (2010-2020). *Frontiers in Marine Science*, 9:733540, May 2022. `doi:10.3389/fmars.2022.733540`.

[22] Marine Fourrier, Laurent Coppola, Fabrizio D'Ortenzio, Christophe Migon, and Jean-Pierre Gattuso. Impact of intermittent convection in the northwestern mediterranean sea on oxygen content, nutrients, and the carbonate system. *Journal of Geophysical Research: Oceans*, 127(9):e2022JC018615, September 2022. `doi:10.1029/2022JC018615`.

[23] Marco Reale, Gianpiero Cossarini, Paolo Lazzari, Tomas Lovato, Giorgio Bolzon, Simona Masina, Cosimo Solidoro, and Stefano Salon. Acidification, deoxygenation, and nutrient and biomass declines in a warming Mediterranean Sea. *Biogeosciences*, 19(17):4035–4065, September 2022. `doi:10.5194/bg-19-4035-2022`.

[24] Tal Ben-Ezra, Alon Blachinsky, Shiran Gozali, Anat Tsemel, Yotam Fadida, Dan Tchernov, Yoav Lehahn, Tatiana Margo Tsagaraki, Ilana Berman-Frank, and Michael Krom. Interannual changes in nutrient and phytoplankton dynamics in the eastern Mediterranean sea (EMS) predict the consequences of climate change; results from the sdot-yam time-series station 2018-2022, June 2024. `doi:10.1101/2024.06.24.600321`.